# TASK-UNIFORM CONVERGENCE AND BACKWARD TRANSFER IN FEDERATED DOMAIN-INCREMENTAL LEARNING WITH PARTIAL PARTICIPATION

## ABSTRACT

Real-world federated systems seldom operate on static data: input distributions drift while privacy rules forbid raw-data sharing. We study this setting as Federated Domain-Incremental Learning (FDIL), where (i) clients are heterogeneous, (ii) tasks arrive sequentially with shifting domains, yet (iii) the label space remains fixed. Two theoretical pillars remain missing for FDIL under realistic deployment: a guarantee of backward knowledge transfer (BKT) and a convergence rate that holds across the sequence of *all* tasks with *partial participation*. We introduce SPECIAL (Server-Proximal Efficient Continual Aggregation for Learning), a simple, memory-free FDIL algorithm that adds a single server-side "anchor" to vanilla FedAvg: in each round, the server nudges the uniformly sampled participated clients update toward the previous global model with a lightweight proximal term. This anchor curbs cumulative drift without replay buffers, synthetic data, or task-specific heads, keeping communication and model size unchanged. Our theory shows that SPECIAL (i) *preserves earlier tasks:* a BKT bound caps any increase in prior-task loss by a drift-controlled term that shrinks with more rounds, local epochs, and participating clients; and (ii) *learns efficiently across all tasks:* the first communication-efficient non-convex convergence rate for FDIL with partial participation, $\mathcal{O}(\sqrt{E/(NT)})$, with $E$ local epochs, $T$ communication rounds, and $N$ participated clients per round, matching single-task FedAvg while explicitly separating optimization variance from inter-task drift. Experimental results further demonstrate the effectiveness of SPECIAL.

## 1 INTRODUCTION

Modern learning systems increasingly face *temporal non-stationarity*: data arrive as a sequence of tasks whose distributions evolve over time. Continual learning (CL), also called incremental learning (IL), therefore develops algorithms that acquire new task knowledge efficiently, retain earlier performance to mitigate catastrophic forgetting, and exploit forward and backward transfer (Lopez-Paz & Ranzato, 2017; Chen & Liu, 2018). In parallel, Federated Learning (FL) enables collaborative training across clients that hold heterogeneous data without sharing raw samples (McMahan et al., 2017). FL brings its own challenges such as client heterogeneity, communication limits, and privacy rules. When the *temporal* shifts of CL meet the *spatial* heterogeneity of FL, naively applying either paradigm fails: centralized CL violates privacy, while standard FL presumes a stationary distribution and offers no retention guarantees. Federated Continual Learning (FCL) (Yoon et al., 2021) emerges as the principled fusion addressing *both* kinds of non-stationarity.

Most FCL studies implicitly target class-incremental scenarios (FCIL) in which new classes appear. Yet many real deployments exhibit *domain shifts under a fixed label set*, namely the federated domain-incremental learning (FDIL) regime. Consider a hospital consortium that refines a diagnostic model while scanners, protocols, or demographics drift; disease labels stay fixed and raw images remain private. FDIL is therefore not a niche corner case but a practically dominant regime that demands methods tailored to domain drift *and* federated privacy constraints.

Despite a flurry of algorithms such as memory replay (Rebuffi et al., 2017; Hou et al., 2019; Wu et al., 2019), regularization/proximal (Zenke et al., 2017; Li & Hoiem, 2017a), parameter isola-

tion/expansion (Javed & White, 2019), and meta-learning (Finn et al., 2017; Riemer et al., 2019), *two theoretical gaps remain for FDIL under simultaneous client heterogeneity and temporal domain shift*: (i) the absence of a *backward knowledge transfer* (BKT) guarantee that quantifies when training on later tasks improves or at least preserves earlier-task performance without storing past data; and (ii) the lack of *a **global convergence rate** across **all** tasks*, i.e., the learner's final model should converge on *every* task, not just the last, while accounting for stochastic noise, intra-/inter-client variance, and cumulative domain drift. These gaps are amplified in realistic federated deployments where *partial participation* is the norm: each round selects only a subset of clients due to availability and system constraints. To close these theoretical gaps, the primary question we seek to address is

> *Can we design a simple FDIL framework that handles spatial heterogeneity, temporal non-stationarity, and partial participation, yet guarantees BKT and an task-uniform convergence rate?*

We answer this question with SPECIAL (*Server-Proximal Efficient Continual Aggregation for Learning*), a lightweight, memory-free framework for FDIL with partial participation. In each communication round, the server *uniformly samples* $N$ of $M$ clients (without replacement), aggregates only their updates, and then applies a single *server-side proximal anchor*: it blends the aggregated model with the *previous* global model via a small quadratic proximal step. This one-line change to FEDAVG curbs cumulative parameter drift, letting the model adapt to new domains while implicitly preserving earlier knowledge, *without* replay buffers, synthetic data, or task-specific heads, so both communication cost and model size remain unchanged. Unlike replay- or expansion-based FDIL methods (Li et al., 2024c; Qi et al., 2023), SPECIAL keeps the footprint constant; unlike client-side regularizers (Li et al., 2024b), the proximal interaction happens *only at the server*, and our analysis targets continual, *task-uniform* guarantees. We deliver two key theoretical contributions:

• **Backward knowledge transfer bound under partial participation.** After partially or fully training on a new task, the loss on any earlier task is provably no greater than its previous value plus a drift-controlled term that *shrinks with more rounds, more local epochs, and larger per-round participation $N$*. The bound explicitly captures stochastic variance and client heterogeneity.

• **First task-uniform convergence rate for FDIL with partial participation.** We establish, to our best knowledge, the **first** global non-convex convergence guarantee that holds ***simultaneously across all tasks*** in FDIL under partial participation. With suitable local and global learning rates, the expected gradient norm of SPECIAL decays at $\mathcal{O}(\sqrt{E/(NT)})$, where $E$ is the number of local epochs, $N$ the number of participating clients per round, and $T$ the number of rounds. This matches the communication efficiency of FEDAVG on a single stationary task when expressed per participating update, yet it holds for the *entire sequence* of drifting tasks and is independent of the total client pool size $M$. The bound includes an explicit drift term that quantifies how domain shift accumulates and how it can be suppressed by tuning the proximal weight and the round/epoch trade-off.

Together, these results bridge continual-learning retention theory and federated optimization under realistic sampling, demonstrating that a single server-side proximal anchor can simultaneously curb cumulative drift and enable *provably efficient, task-uniform* learning across a long sequence of evolving domains with partial participation.

## 2 PROBLEM FORMULATION

**Notation.** We denote the numbers of clients, tasks, local epochs, communication rounds, and the regularization coefficient by $M$, $K$, $E$, $T$, and $\lambda$ respectively. A complete notation table appears in Table 3 in Appendix B.

### 2.1 FEDERATED DOMAIN-INCREMENTAL LEARNING

In CL setting, a model observes a sequence of tasks $\mathcal{T} = \{\mathcal{T}_1, \ldots, \mathcal{T}_K\}$. Task $\mathcal{T}_i = \{(x_n^i, y_n^i)\}_{n=1}^{N_i}$ contains $N_i$ data samples with inputs $x_n^i \in \mathcal{X}^i$ and labels $y_n^i \in \mathcal{Y}^i$. In domain-incremental learning (DIL), all tasks share the same label space ($\mathcal{Y}^i = \mathcal{Y}$ for every task $i$), while their input domains differ ($\mathcal{X}^i \neq \mathcal{X}^j$ for some $i \neq j$). We extend DIL to a federated setting with a central server and $M$ clients. Client $m$ can access only its own local data for each task. At task $K$, the goal of FDIL is to

find parameters $\theta_K \in \mathbb{R}^d$ that minimize the cumulative loss across all $K$ tasks:

$$\min_{\theta_K \in \mathbb{R}^d} f_{1:K}(\theta_K) = \sum_{i=1}^{K} f_i(\theta_K) = \frac{1}{M} \sum_{i=1}^{K} \sum_{m=1}^{M} f_{i,m}(\theta_K), \tag{1}$$

where $f_{i,m}(\theta) = \mathbb{E}_{(\boldsymbol{x},y) \sim \mathcal{T}_{i,m}}[f_{i,m}(\theta; \boldsymbol{x}, y)]$ is the possibly non-convex loss of client $m$ on task $i$. Figure 1 illustrates the FDIL workflow. Eq. (1) can be decomposed into the loss $f_K$ on the current task $K$ and the cumulative loss $f_{1:K-1}$ on previous tasks:

$$f_{1:K}(\theta_K) = \underbrace{f_K(\theta_K)}_{\text{current}} + \underbrace{f_{1:K-1}(\theta_K)}_{\text{previous}}. \tag{2}$$

Because privacy regulations and storage limits prevent the server or clients from keeping raw data from earlier tasks, $f_{1:K-1}(\theta_K)$ cannot be evaluated exactly. It can, however, be approximated without full access to past data samples. Experience-replay approaches retain a small, representative subset of earlier data to stand in for the missing objective. In our setting, *we instead treat the previous global model $\theta_{K-1}$, the parameter vector obtained after finishing task $K-1$, as a compact summary of prior knowledge.* Although it stores no raw samples, $\theta_{K-1}$ encodes enough information to regularize learning on the new task and protect performance on earlier ones.

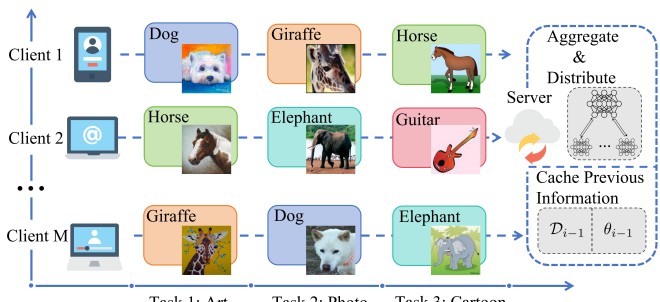

Figure 1: Illustration of FDIL on the PACS dataset. In addition to distribute and aggregate models, the server retains information from the previous task $i-1$, either as a small exemplar buffer $\mathcal{D}_{i-1}$ or, more compactly, as the preceding global model $\theta_{i-1}$.

At the start of task $K$, we initialize the model with the parameters learned on task $K-1$, that is, $\theta_K^0 = \theta_{K-1}$. Inspired by regularization-based continual-learning methods, we add a proximal term to the objective to discourage large departures from $\theta_{K-1}$. This lightweight anchor curbs catastrophic forgetting by keeping optimization centered on the previous solution while still allowing the model to absorb new information from task $K$. Consequently, solving Eq. (1) is equivalent to optimizing the following regularized objective:

$$\theta_K = \arg\min_{\theta \in \mathbb{R}^d} \Big\{ f_{1:K}(\theta) = \underbrace{f_K(\theta)}_{\text{current}} + \underbrace{\lambda \|\theta - \theta_{K-1}\|^2}_{\text{previous}} \Big\}. \tag{3}$$

## 2.2 CLIENT- VS. SERVER-SIDE PROXIMAL TERMS

Most regularization-based approaches keep prior knowledge in play during local training (Li et al., 2020; 2024b; Zhang et al., 2023). In the training process on task $i, i \geq 2$, each client $m$ solves

$$\theta_{i,m} = \arg\min_{\theta \in \mathbb{R}^d} \Big\{ f_{i,m}(\theta) + \lambda \|\theta - \theta_{i-1}\|^2 \Big\}, \tag{4}$$

so every local update is pulled towards the previous global model $\theta_{i-1}$ in each local epoch. While this continual "review" helps preserve earlier knowledge, it can also hinder adaptation to the new task, known as the classic stability-plasticity dilemma (Parisi et al., 2019; Grossberg, 2013).

We illustrate the effect on the first two tasks of Digit-10 (see details in Appendix F). As Figure 2 shows, the client-side proximal term protects task 1 accuracy during the initial rounds of task 2 training but learns task 2 more slowly. In contrast, a server-side proximal term let clients update freely and applies the anchor only at aggregation, striking a better balance: it reaches

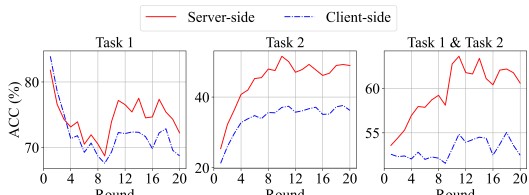

Figure 2: Test accuracy on Task 1, Task 2, and their union for Digit-10 under client-side vs. server-side proximal terms.

higher accuracy on the new task without extra forgetting and even exhibits backward transfers, with task 1 accuracy increasing after 10 rounds on task 2, yielding the best overall performance. This observation motivates our server-side regularization design, which reconciles stability and plasticity more effectively than client-side regularizers (see Section 5 for numerical evaluations).

# 3 THE SPECIAL ALGORITHM

SPECIAL augments vanilla FEDAVG with a single, *server-side proximal anchor* that links each round's update to the global model obtained at the end of the *previous* task. The anchor functions as a compact memory: it curbs catastrophic drift without changing the client workflow or enlarging communication.

**Federated skeleton.** At the start of each communication round for task $i$, the server broadcasts the current global model $\theta_i^t$ to a subset of $N$ out of $M$ clients. We denote the selected index set by $\mathcal{S}_i^t, i \in [K], t \in [T]$. Each selected client downloads the model and performs $E$ epochs of stochastic-gradient decent (SGD) on *only* its own data from task $i$. At the $e$-th epoch, we assume the gradient of client $m$ estimated on the local data sample $\xi_{i,m}^{t,e}$ is an unbiased estimator, which is denoted as $g_{i,m}^{t,e} = \nabla f_{i,m}\left(\theta_{i,m}^{t,e}, \xi_{i,m}^{t,e}\right)$. After local optimization, the client transmits the update vector $\Delta_{i,m}^t = \theta_{i,m}^{t,E} - \theta_i^t$ rather than the full model-back to the server. This vector summarizes all information the client has extracted from its previous data during the round.

After collecting the difference vectors $\Delta_{i,m}^t$ from all participating clients, the

---

**Algorithm 1:** The SPECIAL Algorithm

---

1: Initialize $\theta_0$
2: **for** task $i = 1$ to $K$ **do**
3:    Set task initial point: $\theta_i^0 = \theta_{i-1}$
4:    **for** round $t = 0$ to $T - 1$ **do**
5:      The server randomly selects a set $\mathcal{S}_i^t$ of $N$ clients
6:      **for** client $m \in \mathcal{S}_i^t$ **in parallel do**
7:        Download from server: $\theta_{i,m}^{t,0} = \theta_i^t$
8:        **for** epoch $e = 0$ to $E - 1$ **do**
9:          Calculate: $g_{i,m}^{t,e} = \nabla f_{i,m}\left(\theta_{i,m}^{t,e}, \xi_{i,m}^{t,e}\right)$
10:         Local update: $\theta_{i,m}^{t,e+1} = \theta_{i,m}^{t,e} - \gamma_L g_{i,m}^{t,e}$
11:        **end for**
12:        $\Delta_{i,m}^t = \theta_{i,m}^{t,E} - \theta_{i,m}^{t,0} = -\gamma_L \sum_{e=0}^{E-1} g_{i,m}^{t,e}$
13:        Send $\Delta_{i,m}^t$ to server
14:      **end for**
15:      Server receives all $\Delta_{i,m}^t$
16:      Compute: $\Delta_i^t = \frac{1}{N} \sum_{m \in \mathcal{S}_i^t} \Delta_{i,m}^t$
17:      Intermediate update: $\bar{\theta}_i^{t+1} = \theta_i^t + \gamma_G \Delta_i^t$
18:      Server update according to Eq. (6)
19:    **end for**
20:    Set final model of task $i$: $\theta_i = \theta_i^T$
21: **end for**

---

server computes their average, exactly the aggregation step used by FEDAVG. Applying this average, scaled by a global learning rate $\gamma_G$, yields an intermediate model $\bar{\theta}_i^{t+1} = \theta_i^t + \gamma_G \Delta_i^t$ that reflects what the current task has taught the federation so far, and $\bar{\theta}_i^{t+1}$ is the aggregation result for round $t$ when the current task is the first task of the sequence. Otherwise, the server blends $\bar{\theta}_i^{t+1}$ with the final model from the previous task, $\theta_{i-1}$, via a lightweight proximal step before the round ends (detailed below). The resulting model $\theta_i^{t+1}$ is broadcast at the start of the next round.

**Server-side proximal anchor.** Because clients optimize only the current-task loss, naively applying FEDAVG may rapidly forget features useful to earlier tasks. To counter this, the server solves the quadratic problem

$$\theta_i^{t+1} = \arg\min_{u \in \mathbb{R}^d} \left\{ \left\| u - \bar{\theta}_i^{t+1} \right\|^2 + \lambda \left\| u - \theta_{i-1} \right\|^2 \right\}, \tag{5}$$

whose closed-form solution is

$$\theta_i^{t+1} = \frac{1}{1+\lambda} \bar{\theta}_i^{t+1} + \frac{\lambda}{1+\lambda} \theta_{i-1}. \tag{6}$$

Here, $\bar{\theta}_i^{t+1}$ captures information from the current task, whereas $\theta_{i-1}$ embodies all knowledge accumulated thus far. Their weighted average preserves earlier skills while still adapting to the new domain. The coefficient $\lambda > 0$ is SPECIAL 's *only* new hyper-parameter and governs the stability–plasticity trade-off: $\lambda = 0$ reduces to plain FEDAVG, whereas larger $\lambda$ assigns more weight to retention. The complete procedure is summarized in Algorithm 1.

## 4 THEORETICAL ANALYSIS

We first state the technical assumptions used throughout. We then present two main results: (i) training on a new task can *improve* or at least preserve the loss on every earlier task, and (ii) SPECIAL attains a communication-efficient convergence rate that holds uniformly across *all* tasks, even for non-convex objectives and non-IID data. Complete proofs are deferred to Appendix E.

### 4.1 ASSUMPTION

**Assumption 1** (Bounded Gradients). *For every task $i \in [K]$ and client $m \in [M]$, the stochastic gradient is uniformly bounded, i.e., $\|\nabla f_{i,m}(\theta, \xi)\| \le B$, for $\theta \in \mathbb{R}^d$, $\xi \in \mathcal{T}_{i,m}$, and a constant $B$.*

**Assumption 2** (L-Smooth). *For every task $i \in [K]$, client $m \in [M]$, and all parameter vectors $\theta, \theta' \in \mathbb{R}^d$, the local objective $f_{i,m}(\theta)$ is L-Smooth, i.e., $\|\nabla f_{i,m}(\theta) - \nabla f_{i,m}(\theta')\| \le L \|\theta - \theta'\|$.*

**Assumption 3** (Unbiased Local Gradient estimator). *For every task $i \in [K]$, client $m \in [M]$, and communication round $t \in [T]$, the local stochastic gradient computed on a random sample $\xi_{i,m}^t$ is an unbiased estimate of the true local gradient, i.e., $\mathbb{E}\left[\nabla f_{i,m}\left(\theta_{i,m}^t, \xi_{i,m}^t\right)\right] = \nabla f_{i,m}\left(\theta_{i,m}^t\right)$.*

**Assumption 4** (Bounded Local Gradient Difference). *There exists a constant $\sigma_L > 0$, such that for every task $i \in [K]$, client $m \in [M]$, and communication round $t \in [T]$, the variance of stochastic gradients in client $m$ at task $i$ is bounded by $\mathbb{E}\left\|\nabla f_{i,m}\left(\theta_{i,m}^t, \xi_{i,m}^t\right) - \nabla f_{i,m}\left(\theta_{i,m}^t\right)\right\|^2 \le \sigma_L^2$.*

**Assumption 5** (Bounded Intra-task Gradient Difference). *There exists a constant $\sigma_G > 0$, such that for every task $i \in [K]$, client $m \in [M]$, and communication roud $t \in [T]$, the global variability of local gradient from each client's loss function can be bounded by $\|\nabla f_{i,m}(\theta_i^t) - \nabla f_i(\theta_i^t)\|^2 \le \sigma_G^2$.*

**Assumption 6** (Bounded Inter-task Gradient Difference). *There exists a constant $\sigma_T > 0$, such that for every pair of tasks $i, j \in [K]$ and every parameter vector $\theta \in \mathbb{R}^d$, the difference of their global gradients can be bounded by $\|\nabla f_i(\theta) - \nabla f_j(\theta)\|^2 \le \sigma_T^2$.*

Assumption 1 is routinely used to control magnitude of stochastic gradients in distributed optimization (Lin et al., 2022; Li et al., 2019a; Reddi et al., 2020). Assumption 2 is standard for non-convex analysis. Assumptions 3 and 4 bound, respectively, the sampling bias and variance introduced by per-client minibatches. Assumption 5 is the usual way to quantify *client heterogeneity* in FL. For inter-task drift we additionally assume a bounded gradient discrepancy across tasks (Assumption 6); this *does not* require tasks to be similar. The constant plays the role of a variation/drift budget common in non-stationary optimization and continual learning: our bounds degrade monotonically with the task-drift level, recovering the stationary single-task setting when the drift is zero. This mirrors "positive correlation" or path-length style conditions used to formalize task relatedness in CL (Lin et al., 2022). A formal connection is given in Corollary 2 in Appendix D.

### 4.2 BACKWARD KNOWLEDGE TRANSFER ANALYSIS

We first analyze backward knowledge transfer, namely the phenomenon that training on a new task can *improve* performance on earlier tasks (Benavides-Prado & Riddle, 2022).

**Theorem 1** (BKT under partial participation). *Let $\theta_K^t$ denote the global model after round $t$ of task $K$, and set $\theta_K^0 = \theta_{K-1}$. In each round $\tau$, the server samples a subset $\mathcal{S}_K^\tau \subseteq [M]$ of size $N$ uniformly without replacement and aggregates only those clients. Suppose that for every $\tau \in [t]$, local epoch $e \in [E]$, and client $m \in [M]$, the gradients are $\epsilon$-aligned with the earlier-task gradient at the task-K start, i.e., $\left\langle \nabla f_{1:K-1}\left(\theta_K^0\right), \nabla f_{K,m}\left(\theta_{K,m}^{\tau,e}\right)\right\rangle \ge \epsilon \left\|\nabla f_{1:K-1}\left(\theta_K^0\right)\right\| \cdot \left\|\nabla f_{K,m}\left(\theta_{K,m}^{\tau,e}\right)\right\|, \epsilon \in (0, 1)$, and the local and global learning rates satisfy $\gamma_L \le \frac{2\epsilon \left\|\nabla f_{1:K-1}\left(\theta_K^0\right)\right\|}{BLEt\sqrt{ME \cdot \left(\frac{1}{\lambda^2 + 2\lambda}\right)}}$ and $\gamma_G \le \frac{1}{\sqrt{N}(K-1)}$. Then, for every $t \ge 1$, the server update model $\theta_K^t$ generated by SPECIAL satisfies:*

$$\mathbb{E}_t\left[f_{1:K-1}\left(\theta_K^t\right)\right] \le f_{1:K-1}\left(\theta_K^0\right) + \frac{2\epsilon^2 \sigma_L^2 \left\|\nabla f_{1:K-1}\left(\theta_K^0\right)\right\|^2}{(K-1) tEMNLB^2}, \tag{7}$$

*where the expectation is over client sampling and data stochasticity.*

The bound has two parts. The first term is the *initial sub-optimality* on the earlier tasks at the start of task $K$ (since $\theta_K^0 = \theta_{K-1}$). It reflects how well the model already performed on the earlier tasks before seeing any data from task $K$. The second term is a *vanishing correction* that decays with more rounds $t$, more local epochs $E$, and larger per-round participation $N$. Its magnitude scales with $\|\nabla f_{1:K-1}(\theta_K^0)\|^2$ (harder when the earlier-task gradient is large) and with the local variance $\sigma_L^2$, and it is modulated by the server-side proximal weight $\lambda$ via the step-size constraint. When $N = M$, we recover the full-participation case as a corollary (see Corollary 3 in Appendix D); when $N$ decreases, the $1/N$ factor makes the decay slower, matching intuition under partial participation.

**Remark 1** (Positioning to prior BKT theory). *Lin et al. (2022) establish backward transfer in a centralized continual-learning setting, showing that (under a cosine-alignment condition) training on a new task can reduce earlier-task loss at any epoch. In practice, large distribution gaps at task boundaries often cause an initial spike in the global loss, making such improvements difficult to observe in the first few SGD steps. Our bound in Eq. (7) makes this phenomenon explicit through the local-data variance term $\sigma_L^2$: it decomposes the expected earlier-task loss at $\theta_K^t$ into the starting loss plus a* vanishing correction *that shrinks with more communication rounds, local epochs and with broader effective participation, thereby quantifying when and how fast loss recovery occurs after a sharp shift. This mirrors the structure in Lemma 3.2 of Shi & Wang (2023), which separates the retained loss from a prediction/shift error, but our analysis operates in a* federated *regime with non-IID clients and round-by-round subsampling, and the stabilization mechanism is a* server-side *proximal anchor rather than replay or architectural expansion. The alignment parameter $\epsilon$ in Eq. (7) plays the same role as a cosine-similarity lower bound: when gradients for the new task align better with those of the aggregate of earlier tasks, the vanishing term contracts faster. The constants attached to $\sigma_L^2$ make the dependence on stochasticity and heterogeneity transparent. Taken together, Eq. (7) extends centralized BKT theory to FDIL under realistic sampling and no-memory constraints, and it complements our* task-uniform *convergence result by certifying* loss-space *retention even as the model continues to optimize on the new domain.*

## 4.3 Convergence Across All Tasks

Most prior analyses in FCL establish convergence only on the *final* task (Keshri et al., 2024; Han et al., 2023a). This is insufficient for FDIL, whose goal is to converge *simultaneously* on every task in the sequence. Because tasks can be highly heterogeneous, an algorithm that converges on the last task may still diverge on the earlier ones. We therefore derive guarantees for the *global* objective $f_{1:K} \triangleq \sum_{i=1}^{K} f_i$, ensuring progress across all tasks at once. The argument proceeds in two steps: Lemma 1 bounds within-task drift relative to the task start, and Theorem 2 turns this control into a task-uniform convergence rate under partial participation.

Compared with a centralized setting, convergence in federated setting is complicated by *client drift*: local updates amplify stochastic gradient noise, and repeated rounds can greatly inflate parameter variance. The difficulty is even sharper in FDIL, where each task starts from a checkpoint that already encodes all prior tasks. If the deviation $\|\theta_i^t - \theta_i^0\|^2$ grows unchecked, it slows (or even prevents) convergence on the joint objective. SPECIAL limits this growth through a server-side proximal anchor, enabling the following drift bound.

**Lemma 1** (Uniform within-task drift). *Assume the bounded-gradient condition in Assumption 1. For every task $i \in [K]$ and round $t \in [T]$, the sequence $\{\theta_i^t\}$ produced by SPECIAL satisfies*

$$\mathbb{E}_t \left\| \theta_i^t - \theta_i^0 \right\|^2 \leq \frac{\gamma_G^2 \gamma_L^2 E^2 B^2}{\lambda^2}. \tag{8}$$

The squared distance from the task-initial point is governed jointly by the global and local learning rates ($\gamma_G, \gamma_L$), the number of local epochs ($E$), the gradient bound ($B$), and crucially the proximal weight $\lambda$. A larger $\lambda$ tightens the anchor to $\theta_i^0$ and suppresses drift. The bound is independent of the round $t$ and task $i$, implying a uniform deviation cap across all rounds and tasks.

We are now ready to state a convergence guarantee for the cumulative objective $f_{1:K}$ that holds *simultaneously* for *all* tasks.

**Theorem 2** (Task-uniform convergence of SPECIAL). *Let the local learning rate $\gamma_L$ and global learning rate $\gamma_G$ satisfy $\gamma_G \leq \frac{1}{K-1}$, $\gamma_L \leq \frac{1}{8EL}$ and $\gamma_G \gamma_L \leq \frac{1+\lambda}{3EL}$. In each round, the server samples $N$ of $M$ clients uniformly without replacement and aggregates only their updates. Then the*

*sequence $\{\theta_K^t\}$ generated by* SPECIAL *satisfies:*

$$\min_{t \in [T]} \mathbb{E} \left\| \nabla f_{1:K} \left( \theta_K^t \right) \right\|^2 \leq \frac{f_{1:K}^0 - f_{1:K}^*}{\frac{1 - \frac{1}{K}}{2(1+\lambda)} E \gamma_G \gamma_L T} + \Psi, \tag{9}$$

*where $f_{1:K}^0 \triangleq f_{1:K}(\theta_K^0)$, $f_{1:K}^* \triangleq f_{1:K}(\theta_K^*)$, $\theta_K^*$ is an optimal point for task sequence $1 : K$, the expectation is over client sampling and data stochasticity, and*

$$\begin{aligned}
\Psi = \frac{2}{1 - \frac{1}{K}} &\left[ \left( \frac{\gamma_L^2 E^2 L^2}{\lambda^2} + K + \frac{3\gamma_G \gamma_L (M - N) EKL}{(1+\lambda) N (M - 1)} \right) B^2 + \frac{12\gamma_G \gamma_L (M - N) EKL\sigma_G^2}{(1+\lambda) N (M - 1)} \right. \\
&+ \left( 5\gamma_L^2 KEL^2 + \frac{60\gamma_G \gamma_L^3 (M - N) E^2 KL^3}{(1+\lambda) N (M - 1)} \right) \left( \sigma_L^2 + 6E\sigma_G^2 \right) + \frac{3\gamma_G \gamma_L L\sigma_L^2}{2N (1+\lambda)} \\
&+ \left. \frac{(K - 1)^2 E}{K} \cdot \frac{3\gamma_G \gamma_L L}{1+\lambda} \left( \frac{1}{2} + \frac{4 (M - N)}{N (M - 1)} \right) \sigma_T^2 + \left\| \nabla f_{1:K-1} \left( \theta_K^0 \right) \right\|^2 \right].
\end{aligned}$$

The right-hand side of Eq. (9) consists of a *vanishing* term and a *constant* term $\Psi$. Under the step-size conditions of Theorem 2, the vanishing term scales like $\frac{f_{1:K}^0 - f_{1:K}^*}{c\left(1 - \frac{1}{K}\right) ET}$, for a positive constant $c$. Thus the rate accelerates with more rounds $T$ and more local work $E$, but slows as $K$ grows, reflecting that longer histories require more communication to reach the same stationarity level.

The residual $\Psi$ is scaled by $\frac{2}{1 - \frac{1}{K}}$, which approaches 2 as $K$ increases, and separates five effects: (i) the within-task drift term from Lemma 1, which grows linearly with $K$; (ii) additional client-drift terms induced by partial participation, which vanish as $N$ approaches $M$; (iii) accumulated stochasticity $(\sigma_L^2, \sigma_G^2)$ across $K$ tasks, suggesting $\gamma_L = \mathcal{O}(1/(\sqrt{K}E))$ to keep this component small; (iv) an explicit task-heterogeneity penalty $\sigma_T^2$, amplified roughly by $\frac{(K-1)^2}{K}$, which motivates $\gamma_G = \mathcal{O}(1/(K - 1))$; and (v) the initial gradient norm $\|\nabla f_{1:K-1}(\theta_K^0)\|^2$, which tightens when earlier tasks are already well fit, echoing Theorem 1. Setting $N = M$ recovers the full-participation case as a corollary (see Corollary 4 in Appendix D).

**Remark 2** (Novelty and relation to prior analyses). *Classical partial-participation analyses for FL with non-convex losses (Yang et al., 2021) do not address task-to-task drift and therefore cannot establish convergence on the sum of $f_{1:K}$. Works that quantify task relatedness via gradient inner product (Han et al., 2023b) or sufficient projection/positive correlation (Lin et al., 2022) do not provide a task-uniform convergence rate under non-IID clients and subsampling. Decentralized continual learning results such as Choudhary et al. (2023) make stronger distributional assumptions (e.g., IID across clients) and analyze different communication graphs. Theorem 2 fills this gap for FDIL with partial participation: it delivers a task-uniform stationarity bound in the presence of client heterogeneity and introduces an explicit inter-task drift term $\sigma_T^2$ that reveals how domain shift accumulates with $K$. The server-side proximal anchor is key to controlling within-task deviation and propagating stability across the task sequence without replay or architectural expansion.*

**Corollary 1.** *Let $\gamma_L = \frac{\lambda}{\sqrt{KT}EL}$, $\gamma_G = \frac{\sqrt{NE}}{(K-1)\lambda L}$, and $\left\| \nabla f_{1:K-1} \left( \theta_K^0 \right) \right\|^2 \leq C$, where $C$ is a constant. While training on task $K$, the task-uniform convergence rate of* SPECIAL *satisfies*

$$\min_{t \in [T]} \mathbb{E} \left\| \nabla f_{1:K} \left( \theta_K^t \right) \right\|^2 = \mathcal{O} \left( \sqrt{E/(NT)} \right).$$

**Remark 3** (Trade-off between computation and communication). *The rate improves with larger $T$ and $N$, matching intuition from standard FL, and it highlights the class tension between local computation and communication (Stich, 2018; Li et al., 2019a; Yang et al., 2021). If $E$ is too large, each client over-optimizes its own data and the global iterate $\theta_K^t$ drifts toward the minimizer of the last task $f_K$ instead of the joint objective $f_{1:K}$. If $E$ is too small, many more rounds are required to reach comparable stationarity.*

## 4.4 INTUITIONS AND PROOF SKETCH

We now highlight key ideas and challenges behind the **task-uniform** convergence of SPECIAL. Compared with convergence proofs in FL, the key distinction lies in characterizing the relation

across tasks. Following Assumption 2 and decomposing $\nabla f_{1:K}$, the loss function can be expanded as:

$$\mathbb{E}_t \left[ f_{1:K} \left( \theta_K^{t+1} \right) \right] \leq f_{1:K} \left( \theta_K^t \right) - \frac{\gamma_G \gamma_L E}{1+\lambda} \underbrace{\left\| \nabla f_K \left( \theta_K^t \right) \right\|^2}_{B_1} - \frac{\gamma_G \gamma_L E}{1+\lambda} \underbrace{\left\langle \nabla f_{1:K-1} \left( \theta_K^t \right), \nabla f_K \left( \theta_K^t \right) \right\rangle}_{B_2}$$

$$+ \underbrace{\left\langle \nabla f_{1:K} \left( \theta_K^t \right), \mathbb{E}_t \left[ \theta_K^{t+1} - \theta_K^t \right] + \frac{\gamma_G \gamma_L E}{1+\lambda} \nabla f_K \left( \theta_K^t \right) \right\rangle}_{B_3} + \frac{KL}{2} \underbrace{\mathbb{E}_t \left\| \theta_K^{t+1} - \theta_K^t \right\|^2}_{B_4},$$

where $B_1$ represents the convergence rate of the single task, $B_2$ measures the alignment of $\nabla f_{1:K-1}$ and $\nabla f_K$, and the last two terms characterize the magnitude of the model update in a single communication round. Bellow, we highlight the key differences: 1) **Multiple gradients.** The expansion involves two types of gradients: the single-task gradient $\left\| \nabla f_K \left( \theta_K^t \right) \right\|^2$ and task-uniform gradient $\left\| \nabla f_{1:K} \left( \theta_K^t \right) \right\|^2$, where $\left\| \nabla f_K \left( \theta_K^t \right) \right\|^2$ should be canceled out based on the the relations with $\left\| \nabla f_{1:K} \left( \theta_K^t \right) \right\|^2$ to address the task-uniform convergence rate. 2) **Partial participation.** Terms $B_3$ and $B_4$ both involve the one-round deviation $\theta_K^{t+1} - \theta_K^t$, and the deviation in $B_3$ is equivalent to the deviation under full participation in expectation, while $B_4$ is analyzed under partial participation since the deviation is embedded within the $\ell_2$-norm, so we introduced Lemma 7 and Lemma 8 to bridge the two terms.

Table 1: The ACC and BWT of all baselines and our SPECIAL on four different datasets.

| | Method | Digit-10 | | VLCS | | PACS | | DN4IL | |
|---|---|---|---|---|---|---|---|---|---|
| | | ACC (%) ↑ | BWT (%) ↑ | ACC (%) ↑ | BWT (%) ↑ | ACC (%) ↑ | BWT (%) ↑ | ACC (%) ↑ | BWT (%) ↑ |
| Memory-based | FedCIL | $50.95 \pm 1.66$ | $-27.56 \pm 4.97$ | $48.84 \pm 0.69$ | $-8.88 \pm 3.55$ | $35.11 \pm 0.77$ | $-10.30 \pm 2.08$ | $14.29 \pm 0.41$ | $-23.01 \pm 0.15$ |
| | MFCL | $61.15 \pm 1.43$ | $-19.58 \pm 2.02$ | $42.89 \pm 5.95$ | $-11.03 \pm 1.99$ | $34.68 \pm 1.05$ | $-14.84 \pm 4.21$ | $19.00 \pm 1.06$ | $-21.78 \pm 1.20$ |
| | SR-FDIL | $61.63 \pm 1.47$ | $-36.38 \pm 2.64$ | $48.71 \pm 2.47$ | $-3.22 \pm 7.26$ | $44.92 \pm 0.92$ | $-16.05 \pm 1.58$ | $21.40 \pm 0.25$ | $-33.38 \pm 0.57$ |
| Memory-free | pFedDIL | $46.17 \pm 2.16$ | $-20.21 \pm 0.31$ | $52.44 \pm 1.93$ | $-8.71 \pm 5.87$ | $39.76 \pm 1.98$ | $-21.85 \pm 0.86$ | $12.47 \pm 0.30$ | $\mathbf{-4.63 \pm 0.14}$ |
| | FLwF-2T | $47.39 \pm 5.35$ | $-33.02 \pm 4.94$ | $53.24 \pm 1.43$ | $-0.32 \pm 6.25$ | $43.04 \pm 2.49$ | $\mathbf{-8.12 \pm 5.24}$ | $23.21 \pm 0.49$ | $-16.29 \pm 0.15$ |
| | SPECIAL-C | $50.61 \pm 2.89$ | $\mathbf{-16.42 \pm 0.57}$ | $51.17 \pm 0.44$ | $-8.98 \pm 4.13$ | $43.30 \pm 0.84$ | $-16.40 \pm 1.06$ | $19.88 \pm 0.34$ | $-23.21 \pm 0.43$ |
| | **SPECIAL (ours)** | $\mathbf{62.12 \pm 0.18}$ | $-21.78 \pm 2.94$ | $\mathbf{54.92 \pm 1.60}$ | $\mathbf{-0.11 \pm 2.51}$ | $\mathbf{45.29 \pm 0.92}$ | $-11.66 \pm 1.13$ | $\mathbf{24.30 \pm 0.15}$ | $-19.50 \pm 0.41$ |

## 5 EXPERIMENTS

We now empirically evaluate SPECIAL. Further details about experiments appear in Appendix F.

**Datasets and models.** We consider four domain-shift datasets: Digit-10, VLCS (Torralba & Efros, 2011), PACS (Li et al., 2017), and DN4IL (Gowda et al., 2023) and use the same backbone a ResNet-18 (He et al., 2016).

**Baselines.** We compare SPECIAL with two baseline families. (i) Memory-based methods: FedCIL (Qi et al., 2023), MFCL (Babakniya et al., 2023a), and SR-FDIL (Li et al., 2024c). (ii) Memory-free methods: pFedDIL (Li et al., 2024b), FLwF-2T (Usmanova et al., 2022), and SPECIAL-C (the client-side version of SPECIAL).

**Configurations.** Unless stated otherwise, the number of local epochs $E$ is set to 5 for every task, and the number of communication rounds is set to 20 for Digit-10 and VLCS and 30 for PACS and DN4IL. To create non-IID partitions, we sample client data proportions from a Dirichlet distribution $\mathrm{Dir}(\alpha)$, where smaller $\alpha$ implies stronger heterogeneity. We set $\alpha = 0.1$ for Digit-10, $\alpha = 0.3$ for VLCS and PACS, and $\alpha = 5$ for DN4IL. Local training uses a batch size of 32 and the learning rate of 0.001, while the global learning rate decays with the task index and it is set to $\gamma_G = \frac{1}{i}$, where $i$ denotes the index of the current task. The federation consists of 8 clients for every dataset, and the server randomly selects 4 clients each round. We implement all algorithms in PyTorch on Python 3 with two NVIDIA Quadro RTX 8000 GPUs. All results are averaged over three seeds: 25, 225, and 2025.

**Evaluation metrics.** Like Lin et al. (2022), we report

$$\mathrm{ACC} = \frac{1}{K} \sum_{i=1}^K A_{K,i}, \quad \mathrm{BWT} = \frac{1}{K-1} \sum_{i=1}^{K-1} \left( A_{K,i} - A_{i,i} \right), \tag{10}$$

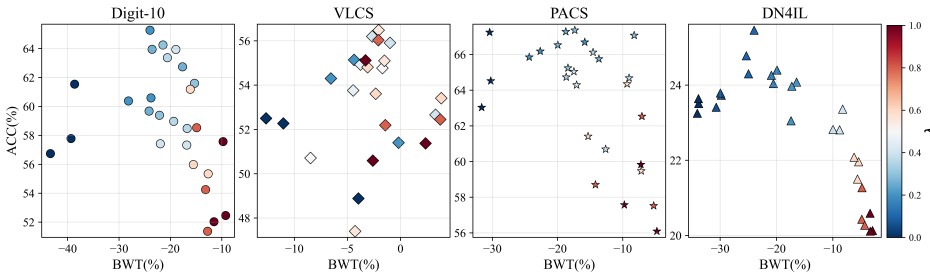

Figure 3: Relation between ACC and BWT under varying values of $\lambda$. Results are demonstrated on three datasets: Digit-10 (circles), VLCS (diamonds), PACS (stars), and DN4IL (triangles), and the color of each point represents the value of $\lambda$ from 0 to 1.

where $A_{i,j}$ is the test accuracy on task $j$ measured with the model obtained after completing task $i$. ACC (average accuracy) reflects overall performance across all learned tasks, while BWT (backward transfer) quantifies how learning new tasks influences accuracy on earlier ones.

**Main results.** Table 1 shows that SPECIAL attains the highest ACC on all three datasets and competitive BWT among *memory-free* methods. Memory-based approaches (e.g., SR-FDIL, FedCIL, and MFCL) can approach SPECIAL on ACC but require storing raw samples or generative synthetic samples. In contrast, SPECIAL is memory-free with only caching the model of the last task. Note that BWT is a *relative* metric, i.e., it depends on both $A_{i,i}$ and $A_{K,i}$ by Eq. (10). Methods with low ACC can still report favorable (less negative) BWT if they underfit current tasks (large $A_{i,i}$, limited improvement in $A_{K,i}$), reflecting stability at the expense of plasticity. SPECIAL achieves strong ACC while keeping BWT at a relatively high level compared with other high-ACC baselines, indicating effective mitigation of forgetting without impeding learning on new domains. Note that SPECIAL performs stably on all three tasks, but other methods lose competitiveness on at least one.

The effect of *where* the proximal term is applied is further confirmed by comparing SPECIAL with its client-side variant, SPECIAL-C. On Digit-10, the dataset with the highest task correlation, "repeated reviewing" during every local update helps preserve earlier knowledge: SPECIAL-C attains slightly lower ACC but higher BWT than server-side SPECIAL, echoing Figure 2. When task correlation weakens, however, the same strategy backfires: continual reviewing cannot boost backward transfer and instead slows adaptation to the new domain, ultimately hurting retention in subsequent tasks. As a result, SPECIAL-C trails SPECIAL on both ACC and BWT for VLCS, PACS, and DN4IL, where the tasks are less related.

**Influence of the proximal weight $\lambda$.** Figure 3 shows how varying $\lambda$ affects performance on three datasets, and the points located closer to the upper-right corner indicate that the corresponding $\lambda$ achieves higher ACC and BWT. Recall that $\lambda$ balances *current-task plasticity* against *past-task stability* when the server combines the intermediate model with the anchor from the previous task. With the increasing of $\lambda$, the BWT keeps growing, which consists with the truth that stability is highly positive correlated with the value of $\lambda$. The extreme high value of $\lambda$ (the red points) causes the worst ACC performance, since the model trained under this $\lambda$ can not learn any new information from the new tasks but only remember the whole information from the first task. Conversely, the extreme low value of $\lambda$ makes the model to be plastic to the new knowledge but forget the previous information which causes the catastrophic forgetting. Therefore, we find that the $\lambda$ values corresponding to higher ACC performance are distributed in the middle of $[0, 1]$, indicating that the ACC is the product of the combination effect of stability and plasticity, the best $\lambda$ should enable the model to learn new information while retaining as much previous information as possible, and this finding is consistent with the discussion about trade-off between backward and forward transfer in Zhang et al. (2023). The optimal $\lambda$ is data-dependent; we use $\lambda = 0.25$ for Digit-10, $\lambda = 0.4$ for VLCS, $\lambda = 0.05$ for PACS, and $\lambda = 0.15$ for DN4IL.

**Ablation study.** Figure 4 explores how communication rounds ($T$), local epochs ($E$), and data-heterogeneity level ($\alpha$) influence performance in Digit-10, thereby testing the claims of Section "Theoretical Analysis". (i) *Communication rounds $T$.* As T grows, ACC rises monotonically—more synchronous updates allow better optimization—while BWT exhibits a U-shape: it is weakest when $T$ is very small (the model leaves a task before converging), becomes increasingly negative as partial convergence accentuates forgetting, and finally improves once $T$ is large enough for the proximal

anchor to propagate useful gradients back to earlier tasks. This behavior matches the trade-off predicted by Theorem 1. (ii) *Local epochs E*. Varying $E$ produces a pattern similar to that for $T$: ACC increases until large $E$ values yield diminishing returns, whereas BWT gradually declines. Longer local training stretches the distance between client updates and the anchor, slowing down the convergence rate in Corollary 1 and thereby reducing backward transfer. (iii) *Dirichlet parameter $\alpha$*. Smaller $\alpha$ (stronger non-IID partitions) degrades ACC, reflecting the additive variance terms $\sigma_L^2$ and $\sigma_G^2$ in Theorem 2. With highly heterogeneous clients, each round introduces greater noise, and more communication is required to attain the same accuracy. Overall, the empirical trends align closely with the theoretical rates: ACC improves with additional communication or computation, while BWT is sensitive to the balance between local plasticity and the stability enforced by the server-side anchor.

Table 2: Evaluation of computational complexity and communication cost of all methods. We report the sum of rounds required to achieve the best performance on each task and the average running time required for each round on all tasks.

| | Method | Digit-10 | | VLCS | | PACS | | DN4IL | |
|---|---|---|---|---|---|---|---|---|---|
| | | Rounds(r) | Time(s) | Rounds(r) | Time(s) | Rounds(r) | Time(s) | Rounds(r) | Time(min) |
| Memory-based | FedCIL | 68 | 523.16 | 65 | 382.20 | 81 | 38.24 | 165 | 17.85 |
| | MFCL | 77 | 969.01 | 71 | 491.46 | 83 | 129.13 | 160 | 38.77 |
| | SR-FDIL | 71 | 357.34 | 62 | 313.21 | 76 | 50.64 | 161 | 18.42 |
| Memory-free | pFedDIL | 76 | 699.43 | 69 | 519.31 | 84 | 75.69 | 167 | 30.71 |
| | FLwF-2T | 67 | 247.90 | 59 | 291.40 | 76 | 43.43 | 156 | 19.33 |
| | SPECIAL-C | 74 | 518.26 | 61 | **283.86** | 75 | 91.98 | 159 | 19.43 |
| | **SPECIAL (ours)** | **63** | **200.66** | **55** | 294.76 | **67** | **26.93** | **145** | **16.23** |

**Computational complexity and communication cost.** Table 2 shows the rounds required for achieving the best performance and the averaging training time cost of each method. The SPECIAL requires the least communication rounds in all four datasets and its averaging training time per round performs the best in three datasets and the second-best on VLCS. Compared with memory-free methods, memory-based methods require additional time to select or generate the data that contains previous information. As for two generative methods, FedCIL deploys the training process of the

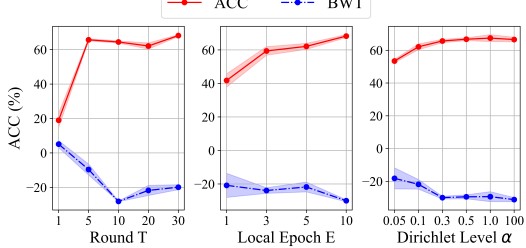

Figure 4: Visualized performance w.r.t $T$, $E$, $\alpha$.

generative model on the client side and MFCL places on the server-side, but this additional training process will increase the computational complexity. Although SR-FDIL does not contain the generative model, it still needs additional process of calculating the scores of each sample to select the optimal subset to be reserved for the future task. Though memory-free methods avoid generating or reserving additional sample, pFedDIL needs to train the additional auxiliary classifier, and all of the memory-free methods except SPECIAL avoid the catastrophic forgetting by adding an additional regularization term on the client-side objective function which increases the difficulty of optimization. In the structure of SPECIAL, the regularizer is deployed on the server-side to aggregate the collected model updates from clients and the optimal model from the previous tasks, so the addition of the regularizer will not influence the whole process, and the computational complexity and communication cost aligns with the FedAvg which is consistent with the Corollary 1.

## LIMITATIONS AND FUTURE WORK

Our analysis assumes bounded gradients/$L$-smoothness and a task-drift bound, and considers *synchronous* partial participation; we do not model stragglers, stale updates, or system constraints. Empirically, we evaluate vision FDIL with ResNet-18; cross-architecture/modality and very large-scale settings are not covered. The proximal weight $\lambda$ is tuned per dataset. Future work: relax assumptions via adaptive step sizes, extend to asynchronous/drop-tolerant aggregation with system metrics, develop self-tuning $\lambda$, and broaden experiments (NLP/time-series/multimodal, larger federated scales, and FCIL/mixed streams).

ETHICS STATEMENT

This research investigates the backward knowledge transfer efficiency and task-uniform convergence rate in Federated Domain Increment Learning settings. By setting a regularizer term on the server side, our algorithm is able to control the deviation of server update to mitigate the catastrophic forgetting. Our framework does not focus on the inherent structure and risks of the deep network. Therefore, our framework should be deployed in conjunction with robust backbone models and modified to adapt to the specific problem and environment.

REPRODUCIBILITY STATEMENT

All experimental parameters are detailed in Section 5 and Appendix F, and all theoretical proofs are listed in Appendix E. The source code and data used for our experiments will be made publicly available upon publication.

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

## A  RELATED WORK

### A.1  CONTINUAL LEARNING

Continual learning aims to train a model on a sequence of tasks under restriction to access to previous data with the goal of mitigating catastrophic forgetting (Wang et al., 2024; Chen & Liu, 2018; Schwarz et al., 2018; Li et al., 2019b). CL methods are typically categorized into two main types: (i) experience replay methods retain or generate synthetic samples from previous tasks to be combined with current task data to be trained (Rebuffi et al., 2017; Wu et al., 2018); (ii) regularization-based methods constrain parameter updates by introducing proximal terms or distillation terms into the loss function to preserver previous information (Li & Hoiem, 2017b; Aich, 2021; Aljundi et al., 2018). From the theoretical perspective, Lin et al. (2022) analyzed backward transfer and inter-task relations with convergence guarantees, while Han et al. (2023a) introduced a decomposition of the loss to separately study overfitting and forgetting under non-convex settings. In this work, we extend CL to the federated setting and build on existing FL theoretical frameworks to establish stronger convergence guarantees for Federated Continual Learning (FCL).

### A.2  FEDERATED CONTINUAL LEARNING

The essence of federated continual learning lies in enabling a model to sequentially learn from distributed tasks across clients while mitigating catastrophic forgetting in a decentralized setting. Studies like Zhou et al. (2022); Li et al. (2024c) adopt experience replay by storing samples with higher importance calculated based on their behaviors during training, but the overhead of calculating and storing sufficient previous samples makes these methods eclipsed compared with generating synthetic data (Babakniya et al., 2023c; Qi et al., 2023; Wuerkaixi et al., 2025), such as TARGET (Zhang et al., 2023), MFCL(Babakniya et al., 2023b) and AF-FCL(Wuerkaixi et al., 2025). Other methods, such as FedWeIT (Yoon et al., 2021) and FLwF-2T (Usmanova et al., 2021), achieve knowledge transfer between tasks via parameter partitioning and knowledge distillation which can be regarded as two main directions of regularization-based methods. However, most FCL methods lack theoretical guarantees. Keshri et al. (2024) extends the proof in CL setting proposed by Han et al. (2023b) to FCL setting, but they research on the convergence rate of loss function only on the last task, while the target of FCL is to ensure convergence across all tasks. In addition, Marfoq et al. (2023) studies convergence under streaming data and provides the theoretical support, but their assumptions are too restrictive. In our work, we establish a solid theoretical guarantee on the convergence rate and performance improvement under reasonable and practically applicable assumptions.

### A.3  FEDERATED DOMAIN-INCREMENTAL LEARNING

Federated Domain-Incremental Learning (FDIL) aims to learn from a sequence of tasks with different domains, which differs from Federated Class-Incremental Learning (FCIL). Existing FDIL methods focus on extracting inter-domain similarities or caching important previous data to mitigate the catastrophic forgetting. Li et al. (2024b) proposed pFedDIL, which selects previous models to be the initial model for the new task based on knowledge intensity and performs knowledge migration by adding the Knowledge Migration (KM) loss. RefFiL (Sun et al., 2024) addresses FDIL by learning domain-invariant knowledge and incorporates domain-specific prompts to alleviate catastrophic forgetting. Similar to ReFed (Li et al., 2024a), SR-FDIL (Li et al., 2024c) enables clients to cache the important samples which can be reuse in the further tasks, and the importance is calculated based on the domain-representative score and cross-client collaborative score. Unlike prior FDIL methods that require significant computational time to prepare for the next arrival task between tasks, our approach transfers knowledge by storing the previous global model, reducing preparation time and preserving privacy.

## B  DETAILED NOTATION

The detailed notation is listed in Table 3.

Table 3: Notations and Terminologies.

| Notation | Description |
|---|---|
| $M$ | Number of all clients |
| $N$ | Number of participated clients per round |
| $T$ | Number of global epochs |
| $E$ | Number of local updates |
| $K$ | Number of tasks |
| $[n]$ | Set of integers $\{1, \ldots, n\}$ |
| $\theta_i$ | Global model after completing training on task $i$ |
| $\theta_i^t$ | Global model at round $t$ in task $i$ |
| $\theta_{i,m}^t$ | Local model on client $m$ at round $t$ in task $i$ |
| $\theta_{i,m}^{t,e}$ | Local model on client $m$ at epoch $e$ of round $t$ in task $i$ |
| $\Delta_{i,m}^t$ | Update vector on client $m$ of round $t$ in task $i$ |
| $f_i$ | Loss function of task $i$ |
| $f_{1:K}$ | Loss function over task 1 to task $K$ |
| $\nabla f_i$ | the gradient of $f_i$ |
| $g_{i,m}^{t,e}$ | Gradient calculated on client m at epoch $e$ of round $t$ in task $i$ |
| $\mathcal{T}$ | Set of all tasks, i.e., $\{\mathcal{T}_1, \ldots, \mathcal{T}_K\}$ |
| $\mathcal{T}_i$ | Dataset of the task $i$ |
| $\mathcal{T}_{i,m}$ | Dataset of client $m$ on task $i$ |
| $\lambda$ | Regularization coefficient |
| $\|\cdot\|^2$ | $\ell_2$-norms |

## C  LEMMA

In the followings, we list a series of lemmas that we use in the theoretical analysis.

**Lemma 2.** *Suppose function $f_1, \ldots, f_K$ are L-smooth, then $f_{1:K} = \sum_{i=1}^K f_i$ is $KL$-smooth.*

**Lemma 3.** *For $x_1, \ldots, x_n \in \mathbb{R}^d$ and $a_1, \ldots, a_n \geq 0$, we have:*

$$\|a_1 x_1 + \cdots + a_n x_n\| \leq \sum_{i=1}^n \|a_i x_i\| = \sum_{i=1}^n a_i \|x_i\|. \tag{11}$$

**Lemma 4.** *For random variables $x_1, \ldots, x_n$, we have:*

$$\mathbb{E} \|x_1 + \cdots + x_n\|^2 \leq n \sum_{i=1}^n \mathbb{E} \|x_i\|^2. \tag{12}$$

**Lemma 5.** *For random variables $x_1, \ldots, x_n$ with $\mathbb{E}[x_i] = 0, \forall i \in [n]$, we have:*

$$\mathbb{E} \|x_1 + \cdots + x_n\|^2 \leq \sum_{i=1}^n \mathbb{E} \|x_i\|^2. \tag{13}$$

**Lemma 6** (Lemma 2 in Yang et al. (2021)). *For any local learning rate satisfying $\gamma_L \leq \frac{1}{8LE}$, we can have the following results:*

$$\frac{1}{M} \sum_{i=1}^M \mathbb{E}\left[\left\|\theta_{i,m}^{t,e} - \theta_i^t\right\|^2\right] \leq 5\gamma_L^2 E\left(\sigma_L^2 + 6E\sigma_G^2\right) + 30\gamma_L^2 E^2 \left\|\nabla f_i\left(\theta_i^t\right)\right\|^2. \tag{14}$$

*Proof.* For any client $m \in [M]$, task $i \in [K]$, and $e \in \{0, \ldots, E-1\}$, we have:

$$
\mathbb{E}\left[\left\|\theta_{i,m}^{t,e} - \theta_i^t\right\|^2\right] = \mathbb{E}\left[\left\|\theta_{i,m}^{t,e-1} - \theta_i^t - \gamma_L g_{i,m}^{t,e-1}\right\|^2\right]
$$

$$
= \mathbb{E}\left[\left\|\theta_{i,m}^{t,e-1} - \theta_i^t - \gamma_L\left(g_{i,m}^{t,e-1} - \nabla f_{i,m}\left(\theta_{i,m}^{t,e-1}\right)\right.\right.\right.
$$

$$
\left.\left.\left. + \nabla f_{i,m}\left(\theta_{i,m}^{t,e-1}\right) - \nabla f_{i,m}\left(\theta_i^t\right) + \nabla f_{i,m}\left(\theta_i^t\right) - \nabla f_i\left(\theta_i^t\right) + \nabla f_i\left(\theta_i^t\right)\right)\right\|^2\right]
$$

$$
\leq \left(1 + \frac{1}{2E-1}\right)\mathbb{E}\left[\left\|\theta_{i,m}^{t,e-1} - \theta_i^t\right\|^2\right] + \gamma_L^2\mathbb{E}\left[\left\|g_{i,m}^{t,e-1} - \nabla f_{i,m}\left(\theta_{i,m}^{t,e-1}\right)\right\|^2\right]
$$

$$
+ 6\gamma_L^2 E \cdot \mathbb{E}\left[\left\|\nabla f_{i,m}\left(\theta_{i,m}^{t,e-1}\right) - \nabla f_{i,m}\left(\theta_i^t\right)\right\|^2 + \left\|\nabla f_{i,m}\left(\theta_i^t\right) - \nabla f_i\left(\theta_i^t\right)\right\|^2 + \left\|\nabla f_i\left(\theta_i^t\right)\right\|^2\right]
$$

$$
\overset{(l_1)}{\leq} \left(1 + \frac{1}{2E-1}\right)\mathbb{E}\left[\left\|\theta_{i,m}^{t,e-1} - \theta_i^t\right\|^2\right] + \gamma_L^2\sigma_L^2
$$

$$
+ 6\gamma_L^2 E\left[L^2\mathbb{E}\left[\left\|\theta_{i,m}^{t,e-1} - \theta_i^t\right\|^2\right] + \sigma_G^2 + \left\|\nabla f_i\left(\theta_i^t\right)\right\|^2\right]
$$

$$
\overset{(l_2)}{=} \left(1 + \frac{1}{2E-1} + 6\gamma_L^2 EL^2\right)\mathbb{E}\left[\left\|\theta_{i,m}^{t,e-1} - \theta_i^t\right\|^2\right] + \gamma_L^2\sigma_L^2 + 6\gamma_L^2 E\sigma_G^2 + 6\gamma_L^2 E\left\|\nabla f_i\left(\theta_i^t\right)\right\|^2
$$

$$
\leq \left(1 + \frac{1}{E-1}\right)\mathbb{E}\left[\left\|\theta_{i,m}^{t,e-1} - \theta_i^t\right\|^2\right] + \gamma_L^2\sigma_L^2 + 6\gamma_L^2 E\sigma_G^2 + 6\gamma_L^2 E\left\|\nabla f_i\left(\theta_i^t\right)\right\|^2,
$$

$$
\tag{15}
$$

where $(l_1)$ follows Assumption 4 and $(l_2)$ is due to Assumption 5.

Unrolling the recursion, we obtain:

$$
\frac{1}{M}\sum_{m=1}^{M}\mathbb{E}\left[\left\|\theta_{i,m}^{t,e} - \theta_i^t\right\|^2\right] \leq \sum_{e=0}^{E-1}\left(1 + \frac{1}{E-1}\right)^e\left[\gamma_L^2\sigma_L^2 + 6\gamma_L^2 E\sigma_G^2 + 6\gamma_L^2 E\left\|\nabla f_i\left(\theta_i^t\right)\right\|^2\right]
$$

$$
\leq (E-1)\left[\left(1 + \frac{1}{E-1}\right)^E - 1\right]\left[\gamma_L^2\sigma_L^2 + 6\gamma_L^2 E\sigma_G^2 + 6\gamma_L^2 E\left\|\nabla f_i\left(\theta_i^t\right)\right\|^2\right]
$$

$$
\overset{(l_3)}{\leq} 5\gamma_L^2 E\left(\sigma_L^2 + 6E\sigma_G^2\right) + 30\gamma_L^2 E^2\left\|\nabla f_i\left(\theta_i^t\right)\right\|^2,
$$

$$
\tag{16}
$$

where $(l_3)$ follows the fact that $\left(1 + \frac{1}{E-1}\right)^E \leq 5$ for $E > 1$. $\qquad\square$

**Lemma 7.** *Assume $\boldsymbol{t}_m \in \mathbb{R}^d, m \in [M], \Lambda \in \mathbb{R}^d$, and $G$ is a constant, then we have:*

$$
\left\|\sum_{m=1}^{M} G\boldsymbol{t}_m + \Lambda\right\|^2 = \left(MG^2 + G\right)\sum_{m=1}^{M}\|\boldsymbol{t}_m\|^2 - \frac{G^2}{2}\sum_{m \neq n}\|\boldsymbol{t}_m - \boldsymbol{t}_n\|^2
$$

$$
+ (1 + MG)\|\Lambda\|^2 - G\sum_{m=1}^{M}\|\boldsymbol{t}_m - \Lambda\|^2.
$$

$$
\tag{17}
$$

*Proof.*

$$
\left\|\sum_{m=1}^{M} G\boldsymbol{t}_m + \Lambda\right\|^2 G^2\sum_{m=1}^{M}\|\boldsymbol{t}_m\|^2 + G^2\sum_{m \neq n}\langle\boldsymbol{t}_m, \boldsymbol{t}_n\rangle + \|\Lambda\|^2 + 2G\sum_{m=1}^{M}\langle\boldsymbol{t}_m, \Lambda\rangle
$$

$$
\overset{(l_4)}{=} G^2\sum_{m=1}^{M}\|\boldsymbol{t}_m\|^2 + G^2\sum_{m \neq n}\frac{1}{2}\left[\|\boldsymbol{t}_m\|^2 + \|\boldsymbol{t}_n\|^2 - \|\boldsymbol{t}_m - \boldsymbol{t}_n\|^2\right]
$$

$$+ \|\Lambda\|^2 + G \sum_{m=1}^{M} \left[ \|\boldsymbol{t}_m\|^2 + \|\Lambda\|^2 - \|\boldsymbol{t}_m - \Lambda\|^2 \right]$$

$$= \left( MG^2 + G \right) \sum_{m=1}^{M} \|\boldsymbol{t}_m\|^2 - \frac{G^2}{2} \sum_{m \neq n} \|\boldsymbol{t}_m - \boldsymbol{t}_n\|^2$$

$$+ (1 + MG) \|\Lambda\|^2 - G \sum_{m=1}^{M} \|\boldsymbol{t}_m - \Lambda\|^2 , \tag{18}$$

where $(l_4)$ follows that $\|\boldsymbol{x} - \boldsymbol{y}\|^2 = \frac{1}{2} \left[ \|\boldsymbol{x}\|^2 + \|\boldsymbol{y}\|^2 - \|\boldsymbol{x} - \boldsymbol{y}\|^2 \right]$. $\qquad \square$

**Lemma 8.** *Assume $\boldsymbol{t}_m \in \mathbb{R}^d, m \in [M], \Lambda \in \mathbb{R}^d$, $G$ is a constant. If we randomly select $N$ clients from all $M$ clients, and the $\mathcal{S}$ is the set of the client index, then we have:*

$$\left\| \sum_{m=1}^{M} G \mathbb{P}\{m \in \mathcal{S}\} \boldsymbol{t}_m + \Lambda \right\|^2 = \left( \frac{N^2 G^2}{M} + \frac{NG}{M} \right) \sum_{m=1}^{M} \|\boldsymbol{t}_m\|^2 - \frac{N(N-1)G^2}{2M(M-1)} \sum_{m \neq n} \|\boldsymbol{t}_m - \boldsymbol{t}_n\|^2$$

$$+ (1 + NG) \|\Lambda\|^2 - \frac{NG}{M} \sum_{m=1}^{M} \|\boldsymbol{t}_m - \Lambda\|^2 . \tag{19}$$

*Proof.*

$$\left\| \sum_{m=1}^{M} G \mathbb{P}\{m \in \mathcal{S}\} \boldsymbol{t}_m + \Lambda \right\|^2$$

$$= G^2 \sum_{m=1}^{M} P\{m \in \mathcal{S}\} \|\boldsymbol{t}_m\|^2 + G^2 \sum_{m \neq n} P\{m, n \in \mathcal{S}\} \langle \boldsymbol{t}_m, \boldsymbol{t}_n \rangle + \|\Lambda\|^2$$

$$+ 2G \sum_{m=1}^{M} P\{m \in \mathcal{S}\} \langle \boldsymbol{t}_m, \Lambda \rangle$$

$$= \frac{NG^2}{M} \sum_{m=1}^{M} \|\boldsymbol{t}_m\|^2 + \frac{N(N-1)G^2}{M(M-1)} \sum_{m \neq n} \langle \boldsymbol{t}_m, \boldsymbol{t}_n \rangle + \|\Lambda\|^2 + \frac{2NG}{M} \sum_{m=1}^{M} \langle \boldsymbol{t}_m, \Lambda \rangle$$

$$\overset{(l_5)}{=} \frac{NG^2}{M} \sum_{m=1}^{M} \|\boldsymbol{t}_m\|^2 + \frac{N(N-1)G^2}{M(M-1)} \sum_{m \neq n} \frac{1}{2} \left[ \|\boldsymbol{t}_m\|^2 + \|\boldsymbol{t}_n\|^2 - \|\boldsymbol{t}_m - \boldsymbol{t}_n\|^2 \right] + \|\Lambda\|^2 \tag{20}$$

$$+ \frac{NG}{M} \sum_{m=1}^{M} \left[ \|\boldsymbol{t}_m\|^2 + \|\Lambda\|^2 - \|\boldsymbol{t}_m - \Lambda\|^2 \right]$$

$$= \left( \frac{N^2 G^2}{M} + \frac{NG}{M} \right) \sum_{m=1}^{M} \|\boldsymbol{t}_m\|^2 - \frac{N(N-1)G^2}{2M(M-1)} \sum_{m \neq n} \|\boldsymbol{t}_m - \boldsymbol{t}_n\|^2$$

$$+ (1 + NG) \|\Lambda\|^2 - \frac{NG}{M} \sum_{m=1}^{M} \|\boldsymbol{t}_m - \Lambda\|^2 ,$$

where $(l_5)$ follows that $\|\boldsymbol{x} - \boldsymbol{y}\|^2 = \frac{1}{2} \left[ \|\boldsymbol{x}\|^2 + \|\boldsymbol{y}\|^2 - \|\boldsymbol{x} - \boldsymbol{y}\|^2 \right]$. $\qquad \square$

**Lemma 9.** *Assume $\boldsymbol{t}_m, \boldsymbol{t}_n \in \mathbb{R}^d, m, n \in [M]$, then we have:*

$$\sum_{m \neq n} \|\boldsymbol{t}_m - \boldsymbol{t}_n\|^2 = 2M \sum_{m=1}^{M} \|\boldsymbol{t}_m\|^2 - 2 \left\| \sum_{m=1}^{M} \boldsymbol{t}_m \right\|^2 . \tag{21}$$

*Proof.*

$$\sum_{m \neq n} \|\boldsymbol{t}_m - \boldsymbol{t}_n\|^2$$

$$= \sum_{m \neq n} \|\boldsymbol{t}_m\|^2 + \sum_{m \neq n} \|\boldsymbol{t}_n\|^2 - \sum_{m \neq n} \langle \boldsymbol{t}_m, \boldsymbol{t}_n \rangle$$

$$= 2(M-1) \sum_{m=1}^{M} \|\boldsymbol{t}_m\|^2 - 2 \left( \sum_{m,n} \langle \boldsymbol{t}_m, \boldsymbol{t}_n \rangle - \sum_{m=1}^{M} \langle \boldsymbol{t}_m, \boldsymbol{t}_m \rangle \right)$$

$$= 2(M-1) \sum_{m=1}^{M} \|\boldsymbol{t}_m\|^2 - 2 \left( \left\langle \sum_{m=1}^{M} \boldsymbol{t}_m, \sum_{n=1}^{M} \boldsymbol{t}_n \right\rangle - \sum_{m=1}^{M} \|\boldsymbol{t}_m\|^2 \right) \tag{22}$$

$$= 2(M-1) \sum_{m=1}^{M} \|\boldsymbol{t}_m\|^2 - 2 \left\| \sum_{m=1}^{M} \boldsymbol{t}_m \right\|^2 + 2 \sum_{m=1}^{M} \|\boldsymbol{t}_m\|^2$$

$$= 2M \sum_{m=1}^{M} \|\boldsymbol{t}_m\|^2 - 2 \left\| \sum_{m=1}^{M} \boldsymbol{t}_m \right\|^2.$$

$\square$

# D  COROLLARY OF THE ASSUMPTION

**Corollary 2.** *Let Assumption 1 and 6 hold. Then, in the case of positive correlation, i.e.,* $\langle \nabla f_i(\theta), \nabla f_j(\theta) \rangle \geq \varepsilon \|\nabla f_i(\theta)\| \cdot \|\nabla f_j(\theta)\|,$ *for any* $\varepsilon \in (0,1)$ *it follows that* $\sigma_T^2 \leq (3 - 2\varepsilon) B^2$.

*Proof.* We have

$$\sigma_T^2 = \|\nabla f_i(\theta) - \nabla f_j(\theta)\|^2$$

$$= \|\nabla f_i(\theta)\|^2 + \|\nabla f_j(\theta)\|^2 - 2 \langle \nabla f_i(\theta), f_j(\theta) \rangle$$

$$\leq \|\nabla f_i(\theta)\|^2 + \|\nabla f_j(\theta)\|^2 - 2\varepsilon \|\nabla f_i(\theta)\| \cdot \|\nabla f_j(\theta)\|$$

$$= (\|\nabla f_i(\theta)\| - \varepsilon \|\nabla f_j(\theta)\|)^2 + (1 - \varepsilon^2) \|\nabla f_j(\theta)\|^2$$

$$\leq \max\{B^2, 2(1-\varepsilon) B^2\}.$$

$\square$

**Corollary 3** (BKT under partial participation). *Let* $\theta_K^t$ *denote the global model after round* $t$ *of task* $K$, *and set* $\theta_K^0 = \theta_{K-1}$. *In each round* $\tau$, *the server samples a subset* $\mathcal{S}_K^\tau \subseteq [M]$ *of size* $N$ *uniformly without replacement and aggregates only those clients. Suppose that for every* $\tau \in [t]$, *local epoch* $e \in [E]$, *and client* $m \in [M]$, *the gradients are* $\epsilon$-*aligned with the earlier-task gradient at the task-* $K$ *start, i.e.,* $\left\langle \nabla f_{1:K-1}(\theta_K^0), \nabla f_{K,m}(\theta_{K,m}^{\tau,e}) \right\rangle \geq \epsilon \left\| \nabla f_{1:K-1}(\theta_K^0) \right\| \cdot \left\| \nabla f_{K,m}(\theta_{K,m}^{\tau,e}) \right\|, \epsilon \in (0,1)$, *and the local and global learning rates satisfy* $\gamma_L \leq \frac{2\epsilon \|\nabla f_{1:K-1}(\theta_K^0)\|}{BLEt \sqrt{ME \cdot \left( \frac{1}{\lambda^2 + 2\lambda} \right)}}$ *and* $\gamma_G \leq \frac{1}{M(K-1)}$.

*Under full participation, for every* $t \geq 1$, *the server update model* $\theta_K^t$ *generated by* SPECIAL *(Algorithm 1) satisfies:*

$$\mathbb{E}_t \left[ f_{1:K-1}(\theta_K^t) \right] \leq f_{1:K-1}(\theta_K^0) + \frac{2\epsilon^2 \sigma_L^2 \left\| \nabla f_{1:K-1}(\theta_K^0) \right\|^2}{(K-1) tEM^2 LB^2}, \tag{23}$$

*where the expectation is over client sampling and data stochasticity.*

**Corollary 4** (Task-uniform *convergence of* SPECIAL *under full participation*). *Let the local learning rate* $\gamma_L$ *and global learning rate* $\gamma_G$ *satisfy* $\gamma_G \leq \frac{1}{K-1}$, $\gamma_L \leq \frac{1}{8EL}$ *and* $\gamma_G \gamma_L \leq \frac{1+\lambda}{3EL}$. *Under full participation, the sequence* $\{\theta_K^t\}$ *generated by* SPECIAL *satisfies:*

$$\min_{t \in [T]} \mathbb{E} \left\| \nabla f_{1:K}(\theta_K^t) \right\|^2 \leq \frac{f_{1:K}^0 - f_{1:K}^*}{\frac{1 - \frac{1}{K}}{2(1+\lambda)} E \gamma_G \gamma_L T} + \Psi', \tag{24}$$

*where*

$$\Psi' = \frac{2}{1 - \frac{1}{K}} \left[ \left( \frac{\gamma_L^2 E^2 L^2}{\lambda^2} + K \right) B^2 + 5\gamma_L^2 KEL^2 \left( \sigma_L^2 + 6E\sigma_G^2 \right) \right.$$

$$\left. + \frac{3\gamma_G \gamma_L L \sigma_L^2}{2M(1+\lambda)} + \frac{(K-1)^2 E}{K} \cdot \frac{3\gamma_G \gamma_L L \sigma_T^2}{2(1+\lambda)} + \left\| \nabla f_{1:K-1} \left( \theta_K^0 \right) \right\|^2 \right].$$

# E COMPLETE PROOFS

## E.1 PROOF OF THEOREM 1

*Proof.* According to the server update rule, we have:

$$\begin{aligned}
\theta_i^{t+1} - \theta_i^t &= \frac{\bar{\theta}_i^{t+1} + \lambda\theta_i^0}{1+\lambda} - \theta_i^t \\
&= \frac{\theta_i^t + \gamma_g \Delta_i^t + \lambda\theta_i^0 - (1+\lambda)\theta_i^t}{1+\lambda} \\
&= \frac{\gamma_G \Delta_i^t}{1+\lambda} + \frac{\lambda\left(\theta_i^0 - \theta_i^t\right)}{1+\lambda}.
\end{aligned} \tag{25}$$

Then, we can obtain:

$$\begin{aligned}
\theta_i^{t+1} - \theta_i^0 &= \theta_i^{t+1} - \theta_i^t - \left(\theta_i^0 - \theta_K^t\right)\theta_i^0 \\
&= \frac{\gamma_G \Delta_i^t}{1+\lambda} + \frac{\lambda\left(\theta_i^0 - \theta_i^t\right) - (1+\lambda)\left(\theta_i^0 - \theta_i^t\right)}{1+\lambda} \\
&= \frac{\gamma_G \Delta_i^t}{1+\lambda} + \frac{\left(\theta_i^t - \theta_i^0\right)}{1+\lambda}.
\end{aligned} \tag{26}$$

Unrolling the recursion, we get:

$$\theta_i^t - \theta_i^0 = \sum_{\tau=0}^{t-1} \left(\frac{1}{1+\lambda}\right)^{\tau+1} \gamma_G \Delta_i^\tau. \tag{27}$$

Then we can expand $f_{1:K-1}\left(\theta_K^0\right)$ by Lemma 2, we have:

$$\begin{aligned}
\mathbb{E}_t &\left[ f_{1:K-1}\left(\theta_K^t\right) \right] \le f_{1:K-1}\left(\theta_K^0\right) + \left\langle \nabla f_{1:K-1}\left(\theta_K^0\right), \mathbb{E}_t\left[\theta_K^t - \theta_K^0\right]\right\rangle \\
&+ \frac{(K-1)}{2} L \mathbb{E}_t \left\| \theta_K^t - \theta_K^0 \right\|^2 \\
\le &f_{1:K-1}\left(\theta_K^0\right) - \gamma_G\gamma_L \left\langle \nabla f_{1:K-1}\left(\theta_K^0\right), \mathbb{E}_t\left[\frac{1}{N}\sum_{\tau=0}^{t-1}\sum_{m\in\mathcal{S}_K^\tau}\sum_{e=0}^{E-1}\left(\frac{1}{1+\lambda}\right)^{\tau+1}\left[g_{K,m}^{\tau,e}\right]\right]\right\rangle \\
&+ \frac{(K-1)}{2} L \cdot \mathbb{E}_t \left\| \frac{-\gamma_G\gamma_L}{N}\sum_{\tau=0}^{t-1}\left(\frac{1}{1+\lambda}\right)^{\tau+1}\sum_{m\in\mathcal{S}_K^\tau}\sum_{e=0}^{E-1}g_{K,m}^{\tau,e} \right\|^2 \\
\le &f_{1:K-1}\left(\theta_K^0\right) \\
&- \frac{\gamma_G\gamma_L}{N}\left\langle \nabla f_{1:K-1}\left(\theta_K^0\right), \sum_{\tau=0}^{t-1}\sum_{m=1}^{M}\mathbb{P}\{m\in\mathcal{S}_K^\tau\}\sum_{e=0}^{E-1}\left(\frac{1}{1+\lambda}\right)^{\tau+1}\nabla f_{K,m}\left(\theta_{K,m}^{\tau,e}\right)\right\rangle \\
&+ \frac{(K-1)}{2} L \cdot \mathbb{E}_t \left\| \frac{-\gamma_G\gamma_L}{N}\sum_{\tau=0}^{t-1}\left(\frac{1}{1+\lambda}\right)^{\tau+1}\sum_{m=1}^{M}\mathbb{P}\{m\in\mathcal{S}_K^\tau\}\sum_{e=0}^{E-1}g_{K,m}^{\tau,e} \right\|^2 \\
\le &f_{1:K-1}\left(\theta_K^0\right) - \frac{\gamma_G\gamma_L}{M}\left\langle \nabla f_{1:K-1}\left(\theta_K^0\right), \sum_{\tau=0}^{t-1}\sum_{m=1}^{M}\sum_{e=0}^{E-1}\left(\frac{1}{1+\lambda}\right)^{\tau+1}\nabla f_{K,m}\left(\theta_{K,m}^{\tau,e}\right)\right\rangle
\end{aligned}$$

$$+ \frac{(K-1)\gamma_G^2\gamma_L^2}{2}L \cdot \underbrace{\mathbb{E}_t \left\| \frac{-1}{N}\sum_{\tau=0}^{t-1}\left(\frac{1}{1+\lambda}\right)^{\tau+1}\sum_{m=1}^{M}\mathbb{P}\{m \in \mathcal{S}_K^\tau\}\sum_{e=0}^{E-1}g_{K,m}^{\tau,e}\right\|^2}_{T_1}. \tag{28}$$

Based on Assumption 3 and the Lemma 8 with $G = -\frac{1}{N}, \boldsymbol{t}_m = \sum_{\tau=0}^{t-1}\left(\frac{1}{1+\lambda}\right)^{\tau+1}\sum_{e=0}^{E-1}g_{K,m}^{\tau,e}$, and $\Lambda = 0$, we can bound term $T_1$ as:

$$\mathbb{E}_t \left\| -\frac{1}{N}\sum_{\tau=0}^{t-1}\left(\frac{1}{1+\lambda}\right)^{\tau+1}\sum_{m=1}^{M}\mathbb{P}\{m \in \mathcal{S}_K^\tau\}\sum_{e=0}^{E-1}g_{K,m}^{\tau,e}\right\|^2$$

$$=\mathbb{E}_t \left\| \sum_{m=1}^{M}-\frac{1}{N}\mathbb{P}\{m \in \mathcal{S}_K^\tau\}\boldsymbol{t}_m\right\|^2$$

$$=-\frac{(N-1)}{2MN(M-1)}\sum_{m \neq n}\mathbb{E}_t\|\boldsymbol{t}_m - \boldsymbol{t}_n\|^2 + \frac{1}{M}\sum_{m=1}^{M}\mathbb{E}_t\|\boldsymbol{t}_m\|^2$$

$$\overset{(a_1)}{=}\frac{1}{M}\sum_{m=1}^{M}\mathbb{E}_t\|\boldsymbol{t}_m\|^2 - \frac{(N-1)}{2MN(M-1)}\left(2M\sum_{m=1}^{M}\mathbb{E}_t\|\boldsymbol{t}_m\|^2 - 2\mathbb{E}_t\left\|\sum_{m=1}^{M}\boldsymbol{t}_m\right\|^2\right) \tag{29}$$

$$=\left(\frac{1}{M}-\frac{(N-1)}{N(M-1)}\right)\sum_{m=1}^{M}\mathbb{E}_t\|\boldsymbol{t}_m\|^2 + \frac{(N-1)}{MN(M-1)}\mathbb{E}_t\left\|\sum_{m=1}^{M}\boldsymbol{t}_m\right\|^2$$

$$\overset{(a_2)}{\leq}\left(\frac{M-N}{MN(M-1)}+\frac{N-1}{N(M-1)}\right)\sum_{m=1}^{M}\mathbb{E}_t\|\boldsymbol{t}_m\|^2$$

$$=\frac{1}{M}\sum_{m=1}^{M}\mathbb{E}_t\|\boldsymbol{t}_m\|^2,$$

where $(a_1)$ is due to Lemma 9 and $(a_2)$ follows Lemma 4. Substituting (29) back to (28), we have:

$$\mathbb{E}_t\left[f_{1:K-1}\left(\theta_K^t\right)\right] \leq f_{1:K-1}\left(\theta_K^0\right)$$

$$-\frac{\gamma_G\gamma_L}{M}\left\langle \nabla f_{1:K-1}\left(\theta_K^0\right), \sum_{\tau=0}^{t-1}\sum_{m=1}^{M}\sum_{e=0}^{E-1}\left(\frac{1}{1+\lambda}\right)^{\tau+1}\nabla f_{K,m}\left(\theta_{K,m}^{\tau,e}\right)\right\rangle$$

$$+\frac{(K-1)}{2}L \cdot \frac{\gamma_G^2\gamma_L^2 Et}{M}\left[\sum_{\tau=0}^{t-1}\sum_{m=1}^{M}\sum_{e=0}^{E-1}\mathbb{E}_t\left\|\left(\frac{1}{1+\lambda}\right)^{\tau+1}g_{K,m}^{\tau,e}\right\|^2\right]$$

$$\overset{(a_3)}{\leq}f_{1:K-1}\left(\theta_K^0\right) - \frac{\gamma_G\gamma_L}{M}\cdot\epsilon\left\|\nabla f_{1:K-1}\left(\theta_K^0\right)\right\| \cdot \left[\sum_{\tau=0}^{t-1}\sum_{m=1}^{M}\sum_{e=0}^{E-1}\left\|\left(\frac{1}{1+\lambda}\right)^{\tau+1}\nabla f_{K,m}\left(\theta_{K,m}^{\tau,e}\right)\right\|\right]$$

$$+\underbrace{\frac{(K-1)}{2}L \cdot \frac{\gamma_G^2\gamma_L^2 Et}{M}\left[\sum_{\tau=0}^{t-1}\sum_{m=1}^{M}\sum_{e=0}^{E-1}\mathbb{E}_t\left\|\left(\frac{1}{1+\lambda}\right)^{\tau+1}g_{K,m}^{\tau,e}\right\|^2\right]}_{T_2},$$

$$\tag{30}$$

where $(a_3)$ is due to $\left\langle \nabla f_{1:K-1}\left(\theta_K^0\right), \nabla f_{K,m}\left(\theta_{K,m}^{i,e}\right)\right\rangle \geq \epsilon\left\|\nabla f_{1:K-1}\left(\theta_K^0\right)\right\| \cdot \left\|\nabla f_{K,m}\left(\theta_{K,m}^{i,e}\right)\right\|$.

Term $T_2$ in (30) can be bounded as

$$T_2 \leq \underbrace{\frac{(K-1)}{2} L \cdot \frac{\gamma_G^2 \gamma_L^2 E t}{M} \left[ \sum_{\tau=0}^{t-1} \sum_{m=1}^{M} \sum_{e=0}^{E-1} \mathbb{E}_t \left\| \left( \frac{1}{1+\lambda} \right)^{\tau+1} \nabla f_{K,m} \left( \theta_{K,m}^{\tau,e} \right) \right\|^2 \right]}_{T_{21}}$$

$$+ \underbrace{\frac{(K-1)}{2} L \cdot \frac{\gamma_G^2 \gamma_L^2 E t}{M} \left[ \sum_{\tau=0}^{t-1} \sum_{m=1}^{M} \sum_{e=0}^{E-1} \mathbb{E}_t \left\| \left( \frac{1}{1+\lambda} \right)^{\tau+1} \left( g_{K,m}^{\tau,e} - \nabla f_{K,m} \left( \theta_{K,m}^{\tau,e} \right) \right) \right\|^2 \right]}_{T_{22}} .$$

$$\tag{31}$$

Since $\gamma_L \leq \frac{2\epsilon \left\| \nabla f_{1:K-1} \left( \theta_K^0 \right) \right\|}{BLEt \sqrt{ME \cdot \left( \frac{1}{\lambda^2 + 2\lambda} \right)}}$ and $\gamma_G \leq \frac{1}{\sqrt{N}(K-1)}$, we have:

$$T_{21} = \frac{\gamma_G \gamma_L}{M\sqrt{N}} \cdot (K-1) \gamma_G \cdot \frac{\gamma_L L E t}{2} \left[ \sum_{\tau=0}^{t-1} \sum_{m=1}^{M} \sum_{e=0}^{E-1} \left\| \left( \frac{1}{1+\lambda} \right)^{\tau+1} \nabla f_{K,m} \left( \theta_{K,m}^{\tau,e} \right) \right\|^2 \right]$$

$$\leq \frac{\gamma_G \gamma_L}{M\sqrt{N}} \cdot \frac{\epsilon \left\| \nabla f_{1:K-1} \left( \theta_K^0 \right) \right\|}{B \sqrt{ME \cdot \left( \frac{1}{\lambda^2 + 2\lambda} \right)}} \left[ \sum_{\tau=0}^{t-1} \sum_{m=1}^{M} \sum_{e=0}^{E-1} \left\| \left( \frac{1}{1+\lambda} \right)^{\tau+1} \nabla f_{K,m} \left( \theta_{K,m}^{\tau,e} \right) \right\|^2 \right]$$

$$\overset{(a_4)}{\leq} \frac{\gamma_G \gamma_L}{M\sqrt{N}} \cdot \frac{\epsilon \left\| \nabla f_{1:K-1} \left( \theta_K^0 \right) \right\| \cdot \left[ \sum_{\tau=0}^{t-1} \sum_{m=1}^{M} \sum_{e=0}^{E-1} \left\| \left( \frac{1}{1+\lambda} \right)^{\tau+1} \nabla f_{K,m} \left( \theta_{K,m}^{\tau,e} \right) \right\|^2 \right]}{\sqrt{\sum_{\tau=0}^{t-1} \left\| \left( \frac{1}{(1+\lambda)^2} \right)^{\tau+1} \right\| \cdot \sum_{m=1}^{M} \sum_{e=0}^{E-1} \left\| \nabla f_{K,m} \left( \theta_{K,m}^{\tau,e} \right) \right\|^2}}$$

$$\leq \frac{\gamma_G \gamma_L}{M\sqrt{N}} \cdot \frac{\epsilon \left\| \nabla f_{1:K-1} \left( \theta_K^0 \right) \right\| \cdot \left[ \sum_{\tau=0}^{t-1} \sum_{m=1}^{M} \sum_{e=0}^{E-1} \left\| \left( \frac{1}{1+\lambda} \right)^{\tau+1} \nabla f_{K,m} \left( \theta_{K,m}^{\tau,e} \right) \right\|^2 \right]}{\sqrt{\sum_{\tau=0}^{t-1} \sum_{m=1}^{M} \sum_{e=0}^{E-1} \left\| \left( \frac{1}{1+\lambda} \right)^{\tau+1} \nabla f_{K,m} \left( \theta_{K,m}^{\tau,e} \right) \right\|^2}}$$

$$\leq \frac{\gamma_G \gamma_L}{M\sqrt{N}} \cdot \epsilon \left\| \nabla f_{1:K-1} \left( \theta_K^0 \right) \right\| \sqrt{\sum_{\tau=0}^{t-1} \sum_{m=1}^{M} \sum_{e=0}^{E-1} \left\| \left( \frac{1}{1+\lambda} \right)^{\tau+1} \nabla f_{K,m} \left( \theta_{K,m}^{\tau,e} \right) \right\|^2}$$

$$\leq \frac{\gamma_G \gamma_L}{M\sqrt{N}} \cdot \epsilon \left\| \nabla f_{1:K-1} \left( \theta_K^0 \right) \right\| \left[ \sum_{\tau=0}^{t-1} \sum_{m=1}^{M} \sum_{e=0}^{E-1} \left\| \left( \frac{1}{1+\lambda} \right)^{\tau+1} \nabla f_{K,m} \left( \theta_{K,m}^{\tau,e} \right) \right\| \right]$$

$$\leq \frac{\gamma_G \gamma_L}{M} \cdot \epsilon \left\| \nabla f_{1:K-1} \left( \theta_K^0 \right) \right\| \left[ \sum_{\tau=0}^{t-1} \sum_{m=1}^{M} \sum_{e=0}^{E-1} \left\| \left( \frac{1}{1+\lambda} \right)^{\tau+1} \nabla f_{K,m} \left( \theta_{K,m}^{\tau,e} \right) \right\| \right],$$

$$\tag{32}$$

where $(a_4)$ follows that

$$\frac{1}{\lambda^2 + 2\lambda} = \frac{1}{(1+\lambda)^2} \cdot \frac{1}{1 - \frac{1}{(1+\lambda)^2}} = \sum_{\tau=0}^{\infty} \left( \frac{1}{(1+\lambda)^2} \right)^{\tau+1} \geq \sum_{\tau=0}^{t-1} \left( \frac{1}{(1+\lambda)^2} \right)^{\tau+1} . \tag{33}$$

Term $T_{22}$ in (31) can be bounded as

$$T_{22} = \frac{(K-1)}{2} L \cdot \frac{\gamma_G^2 \gamma_L^2 E t}{M} \left[ \sum_{\tau=0}^{t-1} \sum_{m=1}^{M} \sum_{e=0}^{E-1} \left( \frac{1}{(1+\lambda)^2} \right)^{\tau+1} \mathbb{E}_t \left\| g_{K,m}^{\tau,e} - \nabla f_{K,m} \left( \theta_{K,m}^{\tau,e} \right) \right\|^2 \right]$$

$$\overset{(a_5)}{\leq} \frac{(K-1)}{2} L \cdot \frac{\gamma_G^2 \gamma_L^2 Et}{M} \left[ \sum_{\tau=0}^{t-1} \left( \frac{1}{(1+\lambda)^2} \right)^{\tau+1} \sum_{m=1}^{M} \sum_{e=0}^{E-1} \sigma_L^2 \right]$$

$$\leq \frac{(K-1)}{2} L \cdot \gamma_G^2 \gamma_L^2 E^2 \sigma_L^2 t \cdot \sum_{\tau=0}^{t-1} \left( \frac{1}{(1+\lambda)^2} \right)^{\tau+1}$$

$$\overset{(a_6)}{\leq} \frac{(K-1)}{2} L \cdot \frac{\gamma_G^2 \gamma_L^2 E^2 \sigma_L^2 t}{\lambda^2 + 2\lambda}$$

$$\overset{(a_7)}{\leq} \frac{2\epsilon^2 \sigma_L^2 \left\| \nabla f_{1:K-1} \left( \theta_K^0 \right) \right\|^2}{(K-1) tEMNLB^2}, \tag{34}$$

where $(a_5)$ is due to Assumption 4, $(a_6)$ follows from (33), and $(a_7)$ holds since $\gamma_L \leq \frac{2\epsilon \left\| \nabla f_{1:K-1} \left( \theta_K^0 \right) \right\|}{BLEt \sqrt{ME \cdot \left( \frac{1}{\lambda^2 + 2\lambda} \right)}}$ and $\gamma_G \leq \frac{1}{\sqrt{N}(K-1)}$.

Plugging (32) and (34) into (34), we have:

$$T_2 \leq \frac{\gamma_G \gamma_L}{M} \cdot \epsilon \left\| \nabla f_{1:K-1} \left( \theta_K^0 \right) \right\| \left[ \sum_{\tau=0}^{t-1} \sum_{m=1}^{M} \sum_{e=0}^{E-1} \left\| \left( \frac{1}{1+\lambda} \right)^{\tau+1} \nabla f_{K,m} \left( \theta_{K,m}^{\tau,e} \right) \right\| \right]$$
$$+ \frac{2\epsilon^2 \sigma_L^2 \left\| \nabla f_{1:K-1} \left( \theta_K^0 \right) \right\|^2}{(K-1) tEMNLB^2}. \tag{35}$$

Plugging (35) back into (30), we have:

$$\mathbb{E}_t \left[ f_{1:K-1} \left( \theta_K^t \right) \right] \leq f_{1:K-1} \left( \theta_K^0 \right) + \frac{2\epsilon^2 \sigma_L^2 \left\| \nabla f_{1:K-1} \left( \theta_K^0 \right) \right\|^2}{(K-1) tEMNLB^2}. \tag{36}$$

$\square$

### E.2 PROOF OF LEMMA 1

*Proof.* According to Eq. 27 and Lemma 3, we have:

$$\mathbb{E}_t \left\| \theta_i^t - \theta_i^0 \right\| = \mathbb{E}_t \left\| \sum_{\tau=0}^{t-1} \left( \frac{1}{1+\lambda} \right)^{\tau+1} \gamma_G \Delta_i^\tau \right\|$$

$$\leq \sum_{\tau=0}^{t-1} \left( \frac{1}{1+\lambda} \right)^{\tau+1} \mathbb{E}_t \left\| \frac{-\gamma_G \gamma_L}{N} \sum_{m \in \mathcal{S}_i^\tau} \sum_{e=0}^{E-1} \nabla f_{i,m} \left( \theta_{i,m}^{\tau,e}, \xi_{i,m}^{\tau,e} \right) \right\|$$

$$\leq \left[ \sum_{\tau=0}^{t-1} \left( \frac{1}{1+\lambda} \right)^{\tau+1} \right] \cdot \mathbb{E}_t \left[ \frac{\gamma_G \gamma_L}{N} \sum_{m \in \mathcal{S}_i^\tau} \sum_{e=0}^{E-1} \left\| \nabla f_{i,m} \left( \theta_{i,m}^{\tau,e}, \xi_{i,m}^{\tau,e} \right) \right\| \right] \tag{37}$$

$$= \left[ \sum_{\tau=0}^{t-1} \left( \frac{1}{1+\lambda} \right)^{\tau+1} \right] \cdot \frac{\gamma_G \gamma_L}{N} \sum_{m=1}^{M} \mathbb{P}\{m \in \mathcal{S}_i^\tau\} \left\| \nabla f_{i,m} \left( \theta_{i,m}^{\tau,e}, \xi_{i,m}^{\tau,e} \right) \right\|$$

$$\overset{(b_1)}{\leq} \frac{1}{\lambda} \cdot \gamma_G \gamma_L EB,$$

where $(b_1)$ follows the truth that $\mathbb{P}\{m \in \mathcal{S}_i^\tau\} = \frac{N}{M}$ Assumption 1 and

$$\sum_{\tau=0}^{t-1} \left( \frac{1}{1+\lambda} \right)^{\tau+1} \leq \sum_{\tau=0}^{\infty} \left( \frac{1}{1+\lambda} \right)^{\tau+1} = \left( \frac{1}{1+\lambda} \right) \cdot \frac{1}{1 - \frac{1}{1+\lambda}} = \frac{1}{\lambda}.$$

Then, we can bound $\left\| \theta_i^t - \theta_i^0 \right\|^2$ as following:

$$\mathbb{E}_t \left\| \theta_i^t - \theta_i^0 \right\|^2 \leq \frac{\gamma_G^2 \gamma_L^2 E^2 B^2}{\lambda^2}. \tag{38}$$

$\square$

### E.3 PROOF OF THEOREM 2

*Proof.* According to Lemma 2, $f_{1:K}$ is $KL$-smooth, and we have the following expansion by taking expectation over the randomness during the training process:

$$
\mathbb{E}_t \left[ f_{1:K} \left( \theta_K^{t+1} \right) \right] \le f_{1:K} \left( \theta_K^t \right) + \left\langle \nabla f_{1:K} \left( \theta_K^t \right), \mathbb{E}_t \left[ \theta_K^{t+1} - \theta_K^t \right] \right\rangle + \frac{KL}{2} \mathbb{E}_t \left\| \theta_K^{t+1} - \theta_K^t \right\|^2
$$

$$
= f_{1:K} \left( \theta_K^t \right) + \left\langle \nabla f_{1:K} \left( \theta_K^t \right), \mathbb{E}_t \left[ \theta_K^{t+1} - \theta_K^t \right] - \frac{\gamma_G \gamma_L E}{1+\lambda} \nabla f_K \left( \theta_K^t \right) + \frac{\gamma_G \gamma_L E}{1+\lambda} \nabla f_K \left( \theta_K^t \right) \right\rangle
$$

$$
+ \frac{KL}{2} \mathbb{E}_t \left\| \theta_K^{t+1} - \theta_K^t \right\|^2
$$

$$
\overset{(c_1)}{=} f_{1:K} \left( \theta_K^t \right) - \frac{\gamma_G \gamma_L E}{1+\lambda} \left\| \nabla f_K \left( \theta_K^t \right) \right\|^2 - \frac{\gamma_G \gamma_L E}{1+\lambda} \underbrace{\left\langle \nabla f_{1:K-1} \left( \theta_K^t \right), \nabla f_K \left( \theta_K^t \right) \right\rangle}_{T_4}
$$

$$
+ \underbrace{\left\langle \nabla f_{1:K} \left( \theta_K^t \right), \mathbb{E}_t \left[ \theta_K^{t+1} - \theta_K^t \right] + \frac{\gamma_G \gamma_L E}{1+\lambda} \nabla f_K \left( \theta_K^t \right) \right\rangle}_{T_5} + \frac{KL}{2} \underbrace{\mathbb{E}_t \left\| \theta_K^{t+1} - \theta_K^t \right\|^2}_{T_6},
$$

$$
\tag{39}
$$

where $(c_1)$ holds due to $f_{1:K} \left( \theta_K^t \right) = f_{1:K-1} \left( \theta_K^t \right) + f_K \left( \theta_K^t \right)$.

Then, term $T_4$ in (39) can be expanded as:

$$
T_4 \overset{(c_2)}{=} \frac{1}{2} \left[ \left\| \nabla f_{1:K} \left( \theta_K^t \right) \right\|^2 - \left\| \nabla f_{1:K-1} \left( \theta_K^t \right) \right\|^2 - \left\| \nabla f_K \left( \theta_K^t \right) \right\|^2 \right]
$$

$$
= \frac{1}{2} \left[ \left\| \nabla f_{1:K} \left( \theta_K^t \right) \right\|^2 - \left\| \nabla f_{1:K-1} \left( \theta_K^t \right) - \nabla f_{1:K-1} \left( \theta_K^0 \right) + \nabla f_{1:K-1} \left( \theta_K^0 \right) \right\|^2 \right.
$$

$$
\left. - \left\| \nabla f_K \left( \theta_K^t \right) \right\|^2 \right]
$$

$$
\ge \frac{1}{2} \left[ \left\| \nabla f_{1:K} \left( \theta_K^t \right) \right\|^2 - 2 \left\| \nabla f_{1:K-1} \left( \theta_K^t \right) - \nabla f_{1:K-1} \left( \theta_K^0 \right) \right\|^2 - 2 \left\| \nabla f_{1:K-1} \left( \theta_K^0 \right) \right\|^2 \right.
$$

$$
\left. - \left\| \nabla f_K \left( \theta_K^t \right) \right\|^2 \right]
$$

$$
\overset{(c_3)}{\ge} \frac{1}{2} \left\| \nabla f_{1:K} \left( \theta_K^t \right) \right\|^2 - \frac{1}{2} \left\| \nabla f_K \left( \theta_K^t \right) \right\|^2 - L^2 (K-1)^2 \left\| \theta_K^t - \theta_K^0 \right\|^2 - \left\| \nabla f_{1:K-1} \left( \theta_K^0 \right) \right\|^2
$$

$$
\overset{(c_4)}{\ge} \frac{1}{2} \left\| \nabla f_{1:K} \left( \theta_K^t \right) \right\|^2 - \frac{1}{2} \left\| \nabla f_K \left( \theta_K^t \right) \right\|^2 - \frac{L^2 (K-1)^2 \gamma_G^2 \gamma_L^2 E^2 B^2}{\lambda^2} - \left\| \nabla f_{1:K-1} \left( \theta_K^0 \right) \right\|^2,
$$

$$
\tag{40}
$$

where $(c_2)$ holds because $\langle x, y \rangle = \frac{1}{2} \left[ \| x + y \|^2 - \| x \|^2 - \| y \|^2 \right]$, $(c_3)$ is due to the fact that $f_{1:K-1} \left( \theta_K^t \right)$ is $(L-1)$-smoothness and Assumption 2, and $(c_4)$ follows Lemma 1.

Since we can expand the right side of term $T_5$ as:

$$
\mathbb{E}_t \left[ \theta_K^{t+1} - \theta_K^t \right] + \frac{\gamma_G \gamma_L E}{1+\lambda} \nabla f_K \left( \theta_K^t \right)
$$

$$
= \left( \frac{1}{1+\lambda} \right) \mathbb{E}_t \left[ \frac{-\gamma_G \gamma_L}{N} \sum_{m \in \mathcal{S}_K^t} \sum_{e=0}^{E-1} g_{K,m}^{t,e} + \lambda \left( \theta_K^0 - \theta_K^t \right) + \gamma_G \gamma_L E \nabla f_K \left( \theta_K^t \right) \right]
$$

$$
= \left( \frac{1}{1+\lambda} \right) \mathbb{E}_t \left[ \frac{-\gamma_G \gamma_L}{N} \cdot \frac{N}{M} \sum_{m=1}^{M} \sum_{e=0}^{E-1} \left( \nabla f_{K,m} \left( \theta_{K,m}^{t,e} \right) - \nabla f_{K,m} \left( \theta_{K,m}^{t,0} \right) \right) + \lambda \left( \theta_K^0 - \theta_K^t \right) \right]
$$

$$
= \left( \frac{1}{1+\lambda} \right) \mathbb{E}_t \left[ \frac{-\gamma_G \gamma_L}{M} \sum_{m=1}^{M} \sum_{e=0}^{E-1} \left( \nabla f_{K,m} \left( \theta_{K,m}^{t,e} \right) - \nabla f_{K,m} \left( \theta_{K,m}^{t,0} \right) \right) + \lambda \left( \theta_K^0 - \theta_K^t \right) \right].
$$

$$
\tag{41}
$$

Then, we can further expand $T_5$ as:

$$T_5 = \left(\frac{1}{1+\lambda}\right)\left\langle \nabla f_{1:K}\left(\theta_K^t\right),\right.$$

$$\left.\mathbb{E}_t\left[\frac{-\gamma_G\gamma_L}{M}\sum_{m=1}^{M}\sum_{e=0}^{E-1}\left(\nabla f_{K,m}\left(\theta_{K,m}^{t,e}\right) - \nabla f_{K,m}\left(\theta_{K,m}^{t,0}\right)\right) + \lambda\left(\theta_K^0 - \theta_K^t\right)\right]\right\rangle$$

$$= \left(\frac{1}{1+\lambda}\right)\left\langle \sqrt{\frac{\gamma_G\gamma_L E}{K}}\nabla f_{1:K}\left(\theta_K^t\right),\right.$$

$$\left.\sqrt{\frac{K}{\gamma_G\gamma_L E}}\mathbb{E}_t\left[\frac{-\gamma_G\gamma_L}{M}\sum_{m=1}^{M}\sum_{e=0}^{E-1}\left(\nabla f_{K,m}\left(\theta_{K,m}^{t,e}\right) - \nabla f_{K,m}\left(\theta_{K,m}^{t,0}\right)\right) + \lambda\left(\theta_K^0 - \theta_K^t\right)\right]\right\rangle$$

$$= \frac{1}{2(1+\lambda)}\left[\frac{\gamma_G\gamma_L E}{K}\left\|\nabla f_{1:K}\left(\theta_K^t\right)\right\|^2\right.$$

$$+ \frac{K}{\gamma_G\gamma_L E}\mathbb{E}_t\left\|\frac{-\gamma_G\gamma_L}{M}\sum_{m=1}^{M}\sum_{e=0}^{E-1}\left(\nabla f_{K,m}\left(\theta_{K,m}^{t,e}\right) - \nabla f_{K,m}\left(\theta_{K,m}^{t,0}\right)\right) + \lambda\left(\theta_K^0 - \theta_K^t\right)\right\|^2$$

$$- \mathbb{E}_t\left\|\sqrt{\frac{K}{\gamma_G\gamma_L E}}\left[\frac{-\gamma_G\gamma_L}{M}\sum_{m=1}^{M}\sum_{e=0}^{E-1}\left(\nabla f_{K,m}\left(\theta_{K,m}^{t,e}\right) - \nabla f_{K,m}\left(\theta_{K,m}^{t,0}\right)\right) + \lambda\left(\theta_K^0 - \theta_K^t\right)\right]\right.$$

$$\left.\left.-\sqrt{\frac{\gamma_G\gamma_L E}{K}}\nabla f_{1:K}\left(\theta_K^t\right)\right\|^2\right]$$

$$= \frac{1}{2(1+\lambda)}\left[\frac{\gamma_G\gamma_L E}{K}\left\|\nabla f_{1:K}\left(\theta_K^t\right)\right\|^2 + T_{51} + T_{52}\right]. \tag{42}$$

Term $T_{51}$ in (42) can be bounded as:

$$T_{51} \leq 2\mathbb{E}_t\left\|\frac{-\gamma_G\gamma_L}{M}\sum_{m=1}^{M}\sum_{e=0}^{E-1}\left(\nabla f_{K,m}\left(\theta_{K,m}^{t,e}\right) - \nabla f_{K,m}\left(\theta_{K,m}^{t,0}\right)\right)\right\|^2 + 2\left\|\lambda\left(\theta_K^0 - \theta_K^t\right)\right\|^2$$

$$\overset{(c_5)}{\leq} \frac{2\gamma_G^2\gamma_L^2 E}{M}\sum_{m=1}^{M}\sum_{e=0}^{E-1}\mathbb{E}_t\left\|\nabla f_{K,m}\left(\theta_{K,m}^{t,e}\right) - \nabla f_{K,m}\left(\theta_{K,m}^{t,0}\right)\right\|^2 + 2\lambda^2\left\|\theta_K^0 - \theta_K^t\right\|^2$$

$$\overset{(c_6)}{\leq} \frac{2\gamma_G^2\gamma_L^2 EL^2}{M}\sum_{m=1}^{M}\sum_{e=0}^{E-1}\mathbb{E}_t\left\|\theta_{K,m}^{t,e} - \theta_{K,m}^{t,0}\right\|^2 + 2\gamma_G^2\gamma_L^2 E^2 B^2$$

$$\overset{(c_7)}{\leq} 2\gamma_G^2\gamma_L^2 E^2 L^2 \cdot \left(5\gamma_L^2 E\left(\sigma_L^2 + 6E\sigma_G^2\right) + 30\gamma_L^2 E^2\left\|\nabla f_K\left(\theta_K^t\right)\right\|^2\right) + 2\gamma_G^2\gamma_L^2 E^2 B^2$$

$$= 10\gamma_G^2\gamma_L^4 E^3 L^2\sigma_L^2 + 60\gamma_G^2\gamma_L^4 E^4 L^2\left(\sigma_G^2 + \left\|\nabla f_K\left(\theta_K^t\right)\right\|^2\right) + 2\gamma_G^2\gamma_L^2 E^2 B^2, \tag{43}$$

where $(c_5)$ is due to Lemma 4, $(c_6)$ holds due to Assumption 2 and Lemma 1, and $(c_7)$ follows Lemma 6. Term $T_{52}$ in (42) can be expanded as:

$$T_{52} = \frac{1}{\gamma_G\gamma_L EK}\mathbb{E}_t\left\|\frac{-\gamma_G\gamma_L K}{M}\sum_{m=1}^{M}\sum_{e=0}^{E-1}\left(\nabla f_{K,m}\left(\theta_{K,m}^{t,e}\right) - \nabla f_{K,m}\left(\theta_{K,m}^{t,0}\right)\right)\right.$$

$$\left.+ \lambda K\left(\theta_K^0 - \theta_K^t\right) - \gamma_G\gamma_L E\nabla f_{1:K}\left(\theta_K^t\right)\right\|^2$$

$$= \frac{\gamma_G\gamma_L}{EK}\mathbb{E}_t\left\|-\frac{1}{M}\sum_{i=1}^{K}\sum_{m=1}^{M}\sum_{e=0}^{E-1}\left(\nabla f_{K,m}\left(\theta_{K,m}^{t,e}\right) - \nabla f_{K,m}\left(\theta_{K,m}^{t,0}\right)\right)\right.$$

$$
+ \frac{\lambda K}{\gamma_G \gamma_L} \left( \theta_K^0 - \theta_K^t \right) - E \sum_{i=1}^{K} f_i \left( \theta_K^t \right) \bigg\|^2
$$

$$
= \frac{\gamma_G \gamma_L}{EK} \mathbb{E}_t \left\| -\frac{1}{M} \sum_{i=1}^{K} \sum_{m=1}^{M} \sum_{e=0}^{E-1} \left( \nabla f_{K,m} \left( \theta_{K,m}^{t,e} \right) - \nabla f_K \left( \theta_K^t \right) + \nabla f_i \left( \theta_K^t \right) \right) \right.
$$

$$
\left. + \frac{\lambda K}{\gamma_G \gamma_L} \left( \theta_K^0 - \theta_K^t \right) \right\|^2 . \tag{44}
$$

Substituting terms $T_{51}$ and $T_{52}$ into (42), we have:

$$
T_5 \leq \frac{\gamma_G \gamma_L E}{2K \left( 1 + \lambda \right)} \left\| \nabla f_{1:K} \left( \theta_K^t \right) \right\|^2 + \frac{5 \gamma_G \gamma_L^3 K E^2 L^2}{\left( 1 + \lambda \right)} \left( \left( \sigma_L^2 + 6 E \sigma_G^2 \right) + 6 E \left\| \nabla f_K \left( \theta_K^t \right) \right\|^2 \right)
$$

$$
+ \frac{\gamma_G \gamma_L K E B^2}{1 + \lambda} - \frac{\gamma_G \gamma_L}{2 \left( 1 + \lambda \right) E K}
$$

$$
\cdot \mathbb{E}_t \left\| -\frac{1}{M} \sum_{i=1}^{K} \sum_{m=1}^{M} \sum_{e=0}^{E-1} \left( \nabla f_{K,m} \left( \theta_{K,m}^{t,e} \right) - \nabla f_K \left( \theta_K^t \right) + \nabla f_i \left( \theta_K^t \right) \right) + \frac{\lambda K}{\gamma_G \gamma_L} \left( \theta_K^0 - \theta_K^t \right) \right\|^2 . \tag{45}
$$

Term $T_6$ in (39) can be expanded as:

$$
T_6 = \mathbb{E}_t \left[ \left\| \left( \frac{-1}{1 + \lambda} \right) \left( \frac{\gamma_G \gamma_L}{N} \sum_{m \in \mathcal{S}_K^t} \sum_{e=0}^{E-1} g_{K,m}^{t,e} + \lambda \left( \theta_K^t - \theta_K^0 \right) \right) \right\|^2 \right]
$$

$$
= \frac{1}{\left( 1 + \lambda \right)^2} \mathbb{E}_t \left\| \frac{1}{K} \left( \frac{-\gamma_G \gamma_L}{N} \sum_{i=1}^{K} \sum_{m \in \mathcal{S}_K^t} \sum_{e=0}^{E-1} g_{K,m}^{t,e} + \lambda K \left( \theta_K^0 - \theta_K^t \right) \right) \right\|^2
$$

$$
= \frac{1}{K^2 \left( 1 + \lambda \right)^2} \mathbb{E}_t \left\| \frac{-\gamma_G \gamma_L}{N} \sum_{i=1}^{K} \sum_{m \in \mathcal{S}_K^t} \sum_{e=0}^{E-1} g_{K,m}^{t,e} + \lambda K \left( \theta_K^0 - \theta_K^t \right) \right\|^2
$$

$$
= \frac{1}{K^2 \left( 1 + \lambda \right)^2} \mathbb{E}_t \left\| \frac{-\gamma_G \gamma_L}{N} \sum_{i=1}^{K} \sum_{m \in \mathcal{S}_K^t} \sum_{e=0}^{E-1} \left( g_{K,m}^{t,e} - \nabla f_{K,m} \left( \theta_{K,m}^{t,e} \right) \right. \right.
$$

$$
\left. \left. + \nabla f_{K,m} \left( \theta_{K,m}^{t,e} \right) - \nabla f_K \left( \theta_K^t \right) + \nabla f_K \left( \theta_K^t \right) - \nabla f_i \left( \theta_K^t \right) + \nabla f_i \left( \theta_K^t \right) \right) + \lambda K \left( \theta_K^0 - \theta_K^t \right) \right\|^2 . \tag{46}
$$

Then, term $T_6$ can be bounded by three terms as:

$$
T_6 \leq \frac{3}{K^2 \left( 1 + \lambda \right)^2} \left( \underbrace{\mathbb{E}_t \left\| -\frac{\gamma_G \gamma_L}{N} \sum_{i=1}^{K} \sum_{m \in \mathcal{S}_K^t} \sum_{e=0}^{E-1} \left( g_{K,m}^{t,e} - \nabla f_{K,m} \left( \theta_{K,m}^{t,e} \right) \right) \right\|^2}_{T_{61}} + \gamma_G^2 \gamma_L^2 \right.
$$

$$
\left. \cdot \underbrace{\mathbb{E}_t \left\| -\frac{1}{N} \sum_{i=1}^{K} \sum_{m \in \mathcal{S}_K^t} \sum_{e=0}^{E-1} \left( \nabla f_{K,m} \left( \theta_{K,m}^{t,e} \right) - \nabla f_K \left( \theta_K^t \right) + \nabla f_i \left( \theta_K^t \right) \right) + \frac{\lambda K}{\gamma_G \gamma_L} \left( \theta_K^0 - \theta_K^t \right) \right\|^2}_{T_{62}} \right.
$$

$$+ \mathbb{E}_t \left\| -\frac{\gamma_G \gamma_L}{N} \sum_{i=1}^{K} \sum_{m \in \mathcal{S}_K^t} \sum_{e=0}^{E-1} \left( \nabla f_K \left( \theta_K^t \right) - \nabla f_i \left( \theta_K^t \right) \right) \right\|^2 \underbrace{\phantom{xxxxxxxxxxxxxxxxxxxxxxxxxxxxxxxxxxxx}}_{T_{63}} \Bigg). \tag{47}$$

Term $T_{61}$ in (47) can be bounded as:

$$T_{61} = \mathbb{E}_t \left\| -\frac{\gamma_G \gamma_L}{N} \sum_{i=1}^{K} \sum_{m=1}^{M} \mathbb{P}\{m \in \mathcal{S}_K^t\} \sum_{e=0}^{E-1} \left( g_{K,m}^{t,e} - \nabla f_{K,m} \left( \theta_{K,m}^{t,e} \right) \right) \right\|^2 \leq \frac{\gamma_G^2 \gamma_L^2 E K \sigma_L^2}{N}. \tag{48}$$

Term $T_{62}$ in (47) can be expanded as:

$$\begin{aligned}
T_{62} = & \mathbb{E}_t \left\| -\frac{1}{N} \sum_{i=1}^{K} \sum_{m=1}^{M} \mathbb{P}\{m \in \mathcal{S}_K^t\} \sum_{e=0}^{E-1} \left( \nabla f_{K,m} \left( \theta_{K,m}^{t,e} \right) - \nabla f_K \left( \theta_K^t \right) + \nabla f_i \left( \theta_K^t \right) \right) \right. \\
& \left. + \frac{\lambda K}{\gamma_G \gamma_L} \left( \theta_K^0 - \theta_K^t \right) \right\|^2.
\end{aligned} \tag{49}$$

Term $T_{63}$ in (47) can be expanded as:

$$\begin{aligned}
T_{63} = & \mathbb{E}_t \left\| -\frac{\gamma_G \gamma_L}{N} \sum_{i=1}^{K} \sum_{m=1}^{M} \mathbb{P}\{m \in \mathcal{S}_K^t\} \sum_{e=0}^{E-1} \left( \nabla f_K \left( \theta_K^t \right) - \nabla f_i \left( \theta_K^t \right) \right) \right\|^2 \\
= & \mathbb{E}_t \left\| -\frac{\gamma_G \gamma_L}{M} \sum_{i=1}^{K} \sum_{m=1}^{M} \sum_{e=0}^{E-1} \left( \nabla f_K \left( \theta_K^t \right) - \nabla f_i \left( \theta_K^t \right) \right) \right\|^2 \\
= & \mathbb{E}_t \left\| -\frac{\gamma_G \gamma_L}{M} \sum_{i=1}^{K-1} \sum_{m=1}^{M} \sum_{e=0}^{E-1} \left( \nabla f_K \left( \theta_K^t \right) - \nabla f_i \left( \theta_K^t \right) \right) \right\|^2 \\
\leq & \gamma_G^2 \gamma_L^2 \left( K - 1 \right)^2 E^2 \sigma_T^2.
\end{aligned} \tag{50}$$

Substituting (48), (49), and (50) back into (47), we have:

$$\begin{aligned}
T_6 \leq & \frac{3 \gamma_G^2 \gamma_L^2 E \sigma_L^2}{NK \left( 1 + \lambda \right)^2} + \frac{3 \gamma_G^2 \gamma_L^2 \left( K - 1 \right)^2 E^2 \sigma_T^2}{K^2 \left( 1 + \lambda \right)^2} + \frac{3 \gamma_G^2 \gamma_L^2}{K^2 \left( 1 + \lambda \right)^2} \\
& \cdot \mathbb{E}_t \left\| -\frac{1}{N} \sum_{i=1}^{K} \sum_{m=1}^{M} \mathbb{P}\{m \in \mathcal{S}_K^t\} \sum_{e=0}^{E-1} \left( \nabla f_{K,m} \left( \theta_{K,m}^{t,e} \right) - \nabla f_K \left( \theta_K^t \right) + \nabla f_i \left( \theta_K^t \right) \right) \right. \\
& \left. + \frac{\lambda K}{\gamma_G \gamma_L} \left( \theta_K^0 - \theta_K^t \right) \right\|^2.
\end{aligned} \tag{51}$$

Substituting (40), (45), and (51) back into (39), we have:

$$\begin{aligned}
\mathbb{E}_t \left[ f_{1:K} \left( \theta_K^{t+1} \right) \right] \leq & f_{1:K} \left( \theta_K^t \right) - \frac{\gamma_G \gamma_L E}{1 + \lambda} \left( \frac{1}{2} - 30 K \gamma_L^2 L^2 E^2 \right) \left\| \nabla f_K \left( \theta_K^t \right) \right\|^2 \\
& - \frac{1}{2} \left( \frac{\gamma_G \gamma_L E}{1 + \lambda} \right) \left( 1 - \frac{1}{K} \right) \left\| \nabla f_{1:K} \left( \theta_K^t \right) \right\|^2 + \frac{\gamma_G \gamma_L E}{1 + \lambda} \left( \frac{L^2 \left( K - 1 \right)^2 \gamma_G^2 \gamma_L^2 E^2}{\lambda^2} + K \right) B^2 \\
& + \frac{\gamma_G \gamma_L E}{1 + \lambda} \left( 5 \gamma_L^2 K E L^2 \left( \sigma_L^2 + 6 E \sigma_G^2 \right) + \frac{3 \gamma_G \gamma_L L}{2 \left( 1 + \lambda \right)} \left( \frac{\sigma_L^2}{N} + \frac{\left( K - 1 \right)^2 E}{K} \sigma_T^2 \right) \right) \\
& + \frac{\gamma_G \gamma_L E}{1 + \lambda} \left\| \nabla f_{1:K-1} \left( \theta_K^0 \right) \right\|^2 + \frac{3 \gamma_G^2 \gamma_L^2 L}{2 K \left( 1 + \lambda \right)^2}
\end{aligned}$$

$$\cdot \mathbb{E}_t \underbrace{\left\| -\frac{1}{N} \sum_{i=1}^{K} \sum_{m=1}^{M} \mathbb{P}\{m \in \mathcal{S}_K^t\} \sum_{e=0}^{E-1} \left( \nabla f_{K,m} \left( \theta_{K,m}^{t,e} \right) - \nabla f_K \left( \theta_K^t \right) + \nabla f_i \left( \theta_K^t \right) \right) + \frac{\lambda K}{\gamma_G \gamma_L} \left( \theta_K^0 - \theta_K^t \right) \right\|^2}_{T_7}$$

$$- \frac{\gamma_G \gamma_L}{2 (1+\lambda) EK}$$

$$\cdot \mathbb{E}_t \underbrace{\left\| -\frac{1}{M} \sum_{i=1}^{K} \sum_{m=1}^{M} \sum_{e=0}^{E-1} \left( \nabla f_{K,m} \left( \theta_{K,m}^{t,e} \right) - \nabla f_K \left( \theta_K^t \right) + \nabla f_i \left( \theta_K^t \right) \right) + \frac{\lambda K}{\gamma_G \gamma_L} \left( \theta_K^0 - \theta_K^t \right) \right\|^2}_{T_8}.$$

$$(52)$$

According to Lemma 8 with $\boldsymbol{t}_m = \sum_{i=1}^{K} \sum_{e=0}^{E-1} \left( \nabla f_{K,m} \left( \theta_{K,m}^{t,e} \right) - \nabla f_K \left( \theta_K^t \right) + \nabla f_i \left( \theta_K^t \right) \right)$, $G = -\frac{1}{N}$, and $\Lambda = \frac{\lambda K}{\gamma_G \gamma_L} \left( \theta_K^0 - \theta_K^t \right)$, we can expand term $T_7$ as:

$$T_7 = \left( \frac{N^2}{MN^2} - \frac{N}{MN} \right) \sum_{m=1}^{M} \mathbb{E}_t \| \boldsymbol{t}_m \|^2 - \frac{N(N-1)}{2MN^2(M-1)} \sum_{m \neq n} \mathbb{E}_t \| \boldsymbol{t}_m - \boldsymbol{t}_n \|^2$$

$$+ \left( 1 - \frac{N}{N} \right) \mathbb{E}_t \| \Lambda \|^2 + \frac{N}{MN} \sum_{m=1}^{M} \mathbb{E}_t \| \boldsymbol{t}_m - \Lambda \|^2 \qquad (53)$$

$$= -\frac{(N-1)}{2MN(M-1)} \sum_{m \neq n} \mathbb{E}_t \| \boldsymbol{t}_m - \boldsymbol{t}_n \|^2 + \frac{1}{M} \sum_{m=1}^{M} \mathbb{E}_t \| \boldsymbol{t}_m - \Lambda \|^2.$$

According to Lemma 7 with $\boldsymbol{t}_m = \sum_{i=1}^{K} \sum_{e=0}^{E-1} \left( \nabla f_{K,m} \left( \theta_{K,m}^{t,e} \right) - \nabla f_K \left( \theta_K^t \right) + \nabla f_i \left( \theta_K^t \right) \right)$, $G = -\frac{1}{M}$, and $\Lambda = \frac{\lambda K}{\gamma_G \gamma_L} \left( \theta_K^0 - \theta_K^t \right)$, we can expand term $T_8$ as:

$$T_8 = \left( \frac{M}{M^2} - \frac{1}{M} \right) \sum_{m=1}^{M} \mathbb{E}_t \mathbb{E}_t \| \boldsymbol{t}_m \|^2 - \frac{1}{2M^2} \sum_{m \neq n} \mathbb{E}_t \| \boldsymbol{t}_m - \boldsymbol{t}_n \|^2$$

$$+ \left( 1 - \frac{M}{M} \right) \| \Lambda \|^2 + \frac{1}{M} \sum_{m=1}^{M} \mathbb{E}_t \| \boldsymbol{t}_m - \Lambda \|^2 \qquad (54)$$

$$= -\frac{1}{2M^2} \sum_{m \neq n} \mathbb{E}_t \| \boldsymbol{t}_m - \boldsymbol{t}_n \|^2 + \frac{1}{M} \sum_{m=1}^{M} \mathbb{E}_t \| \boldsymbol{t}_m - \Lambda \|^2.$$

Since $T_7 \geq 0$ and $T_8 \geq 0$, we have:

$$\begin{cases} T_7 = -\frac{(N-1)}{2MN(M-1)} \sum_{m \neq n} \mathbb{E}_t \| \boldsymbol{t}_m - \boldsymbol{t}_n \|^2 + \frac{1}{M} \sum_{m=1}^{M} \mathbb{E}_t \| \boldsymbol{t}_m - \Lambda \|^2 \geq 0, \\[2mm] T_8 = -\frac{1}{2M^2} \sum_{m \neq n} \mathbb{E}_t \| \boldsymbol{t}_m - \boldsymbol{t}_n \|^2 + \frac{1}{M} \sum_{m=1}^{M} \mathbb{E}_t \| \boldsymbol{t}_m - \Lambda \|^2 \geq 0. \end{cases} \qquad (55)$$

$$\Rightarrow \sum_{m \neq n} \mathbb{E}_t \| \boldsymbol{t}_m - \boldsymbol{t}_n \|^2 \leq 2M \sum_{m=1}^{M} \mathbb{E}_t \| \boldsymbol{t}_m - \Lambda \|^2 \leq 2M \cdot \frac{N(M-1)}{M(N-1)} \sum_{m=1}^{M} \mathbb{E}_t \| \boldsymbol{t}_m - \Lambda \|^2$$

For $\boldsymbol{t}_m$, we have:

$$\sum_{m=1}^{M} \mathbb{E}_t \| \boldsymbol{t}_m \|^2 = \sum_{m=1}^{M} \mathbb{E}_t \left\| \sum_{i=1}^{K} \sum_{e=0}^{E-1} \left( \nabla f_{K,m} \left( \theta_{K,m}^{t,e} \right) - \nabla f_K \left( \theta_K^t \right) + \nabla f_i \left( \theta_K^t \right) \right) \right\|^2$$

$$= \sum_{m=1}^{M} \mathbb{E}_t \left\| \sum_{i=1}^{K} \sum_{e=0}^{E-1} \left( \nabla f_{K,m} \left( \theta_{K,m}^{t,e} \right) - \nabla f_{K,m} \left( \theta_K^t \right) + \nabla f_{K,m} \left( \theta_K^t \right) - \nabla f_K \left( \theta_K^t \right) \right. \right.$$

$$+ \nabla f_i \left( \theta_K^t \right) - \nabla f_K \left( \theta_K^t \right) + \nabla f_K \left( \theta_K^t \right) \Big\|^2$$

$$\leq 4 \sum_{m=1}^{M} \mathbb{E}_t \left\| \sum_{i=1}^{K} \sum_{e=0}^{E-1} \left( \nabla f_{K,m} \left( \theta_{K,m}^{t,e} \right) - \nabla f_{K,m} \left( \theta_K^t \right) \right) \right\|^2$$

$$+ 4 \sum_{m=1}^{M} \mathbb{E}_t \left\| \sum_{i=1}^{K} \sum_{e=0}^{E-1} \left( \nabla f_{K,m} \left( \theta_K^t \right) - \nabla f_K \left( \theta_K^t \right) \right) \right\|^2$$

$$+ 4 \sum_{m=1}^{M} \mathbb{E}_t \left\| \sum_{i=1}^{K} \sum_{e=0}^{E-1} \left( \nabla f_i \left( \theta_K^t \right) - \nabla f_K \left( \theta_K^t \right) \right) \right\|^2 + 4 \sum_{m=1}^{M} \mathbb{E}_t \left\| \sum_{i=1}^{K} \sum_{e=0}^{E-1} \nabla f_K \left( \theta_K^t \right) \right\|^2$$

$$\overset{(c_8)}{\leq} 4EK^2 \sum_{m=1}^{M} \sum_{e=0}^{E-1} \mathbb{E}_t \left\| \nabla f_{K,m} \left( \theta_{K,m}^{t,e} \right) - \nabla f_{K,m} \left( \theta_K^t \right) \right\|^2 + 4ME^2 K^2 \left( \sigma_G^2 + \left\| \nabla f_K \left( \theta_K^t \right) \right\|^2 \right)$$

$$+ 4ME^2 \left( K-1 \right)^2 \sigma_T^2$$

$$\overset{(c_9)}{\leq} 4EK^2 L^2 \sum_{m=1}^{M} \mathbb{E}_t \left\| \theta_{K,m}^{t,e} - \theta_K^t \right\|^2 + 4ME^2 K^2 \left( \sigma_G^2 + \left\| \nabla f_K \left( \theta_K^t \right) \right\|^2 \right) + 4ME^2 \left( K-1 \right)^2 \sigma_T^2$$

$$\overset{(c_{10})}{\leq} 20ME^3 K^2 L^2 \gamma_L^2 \left( \sigma_L^2 + 6E\sigma_G^2 \right) + \left( 120ME^4 L^2 K^2 \gamma_L^2 + 4ME^2 K^2 \right) \left\| \nabla f_K \left( \theta_K^t \right) \right\|^2$$

$$+ 4ME^2 K^2 \sigma_G^2 + 4ME^2 \left( K-1 \right)^2 \sigma_T^2, \tag{56}$$

where $(c_8)$ follows Assumption 5 and Assumption 6, $(c_9)$ is due to Assumption 2, and $(c_{10})$ follows Lemma 1.

Based on (53), (54), (55), and (56), we have:

$$\frac{3\gamma_G^2 \gamma_L^2 L}{2K \left( 1+\lambda \right)^2} T_7 - \frac{\gamma_G \gamma_L}{2 \left( 1+\lambda \right) EK} T_8$$

$$= \frac{3\gamma_G^2 \gamma_L^2 L}{2K \left( 1+\lambda \right)^2} \left( -\frac{\left( N-1 \right)}{2MN \left( M-1 \right)} \sum_{m \neq n} \mathbb{E}_t \left\| \boldsymbol{t}_m - \boldsymbol{t}_n \right\|^2 + \frac{1}{M} \sum_{m=1}^{M} \mathbb{E}_t \left\| \boldsymbol{t}_m - \Lambda \right\|^2 \right)$$

$$- \frac{\gamma_G \gamma_L}{2 \left( 1+\lambda \right) EK} \left( -\frac{1}{2M^2} \sum_{m \neq n} \mathbb{E}_t \left\| \boldsymbol{t}_m - \boldsymbol{t}_n \right\|^2 + \frac{1}{M} \sum_{m=1}^{M} \mathbb{E}_t \left\| \boldsymbol{t}_m - \Lambda \right\|^2 \right) \tag{57}$$

$$= \left( \frac{\gamma_G \gamma_L}{4 \left( 1+\lambda \right) EM^2 K} - \frac{3 \left( N-1 \right) \gamma_G^2 \gamma_L^2 L}{4 \left( 1+\lambda \right)^2 KM \left( M-1 \right) N} \right) \sum_{m \neq n} \mathbb{E}_t \left\| \boldsymbol{t}_m - \boldsymbol{t}_n \right\|^2$$

$$+ \left( \frac{3\gamma_G^2 \gamma_L^2 L}{2MK \left( 1+\lambda \right)^2} - \frac{\gamma_G \gamma_L}{2 \left( 1+\lambda \right) EKM} \right) \sum_{m=1}^{M} \mathbb{E}_t \left\| \boldsymbol{t}_m - \Lambda \right\|^2.$$

Since $\sum_{m \neq n} \left\| \boldsymbol{t}_m - \boldsymbol{t}_n \right\|^2 \leq 2M \sum_{m=1}^{M} \left\| \boldsymbol{t}_m - \Lambda \right\|^2$, we have:

$$\frac{3\gamma_G^2 \gamma_L^2 L}{2K \left( 1+\lambda \right)^2} T_7 - \frac{\gamma_G \gamma_L}{2 \left( 1+\lambda \right) EK} T_8$$

$$\leq \sum_{m=1}^{M} \mathbb{E}_t \left\| \boldsymbol{t}_m - \Lambda \right\|^2 \cdot \left( \frac{\gamma_G \gamma_L}{2 \left( 1+\lambda \right) EMK} - \frac{3 \left( N-1 \right) \gamma_G^2 \gamma_L^2 L}{2 \left( 1+\lambda \right)^2 K \left( M-1 \right) N} \right.$$

$$\left. + \frac{3\gamma_G^2 \gamma_L^2 L}{2MK \left( 1+\lambda \right)^2} - \frac{\gamma_G \gamma_L}{2 \left( 1+\lambda \right) EKM} \right)$$

$$= \left( \frac{3\gamma_G^2 \gamma_L^2 L}{2MK \left( 1+\lambda \right)^2} - \frac{3 \left( N-1 \right) \gamma_G^2 \gamma_L^2 L}{2 \left( 1+\lambda \right)^2 K \left( M-1 \right) N} \right) \sum_{m=1}^{M} \mathbb{E}_t \left\| \boldsymbol{t}_m - \Lambda \right\|^2$$

$$
\leq \frac{3\gamma_G^2 \gamma_L^2}{2K(1+\lambda)^2} \cdot \frac{M-N}{MN(M-1)} \left( 2\sum_{m=1}^{M} \mathbb{E}_t \|\boldsymbol{t}_m\|^2 + 2M\mathbb{E}_t \|\Lambda\|^2 \right)
$$

$$
= \frac{3\gamma_G^2 \gamma_L^2 L}{K(1+\lambda)^2} \cdot \frac{M-N}{MN(M-1)} \left( \sum_{m=1}^{M} \mathbb{E}_t \|\boldsymbol{t}_m\|^2 + M\mathbb{E}_t \|\Lambda\|^2 \right)
$$

$$
\overset{(c_{11})}{\leq} \frac{\gamma_G \gamma_L E}{1+\lambda} \cdot \frac{3\gamma_G \gamma_L (M-N) L}{(1+\lambda) N(M-1)} \cdot \left( 20E^2 K L^2 \gamma_L^2 \left( \sigma_L^2 + 6E\sigma_G^2 \right) \right. \tag{58}
$$

$$
\left. + \left( 120E^3 K L^2 \gamma_L^2 + 4EK \right) \left\| \nabla f_K \left( \theta_K^t \right) \right\|^2 + 4EK\sigma_G^2 + \frac{4E(K-1)^2}{K} \sigma_T^2 + EKB^2 \right), \tag{59}
$$

where $(c_{11})$ is due to (56) and Lemma 1.

Then we have:

$$
\mathbb{E}_t \left[ f_{1:K} \left( \theta_K^{t+1} \right) \right] \leq f_{1:K} \left( \theta_K^t \right) - \frac{1}{2} \left( \frac{\gamma_G \gamma_L E}{1+\lambda} \right) \left( 1 - \frac{1}{K} \right) \left\| \nabla f_{1:K} \left( \theta_K^t \right) \right\|^2 - \frac{\gamma_G \gamma_L E}{1+\lambda}
$$

$$
\cdot \left( \frac{1}{2} - 30K\gamma_L^2 L^2 E^2 - \frac{3\gamma_G \gamma_L (M-N) L}{(1+\lambda) N(M-1)} \cdot \left( 120E^3 L^2 K \gamma_L^2 + 4EK \right) \right) \left\| \nabla f_K \left( \theta_K^t \right) \right\|^2
$$

$$
+ \frac{\gamma_G \gamma_L E}{1+\lambda} \left( \frac{L^2 (K-1)^2 \gamma_G^2 \gamma_L^2 E^2}{\lambda^2} + K + \frac{3\gamma_G \gamma_L (M-N) EKL}{(1+\lambda) N(M-1)} \right) B^2
$$

$$
+ \frac{\gamma_G \gamma_L E}{1+\lambda} \left( \left( 5\gamma_L^2 KEL^2 + \frac{60\gamma_G \gamma_L^3 (M-N) E^2 K L^3}{(1+\lambda) N(M-1)} \right) \left( \sigma_L^2 + 6E\sigma_G^2 \right) + \frac{3\gamma_G \gamma_L \sigma_L^2 L}{2N(1+\lambda)} \right.
$$

$$
\left. + \frac{(K-1)^2 E}{K} \left( \frac{3\gamma_G \gamma_L L}{2(1+\lambda)} + \frac{12\gamma_G \gamma_L (M-N) L}{(1+\lambda) N(M-1)} \right) \sigma_T^2 + \frac{12\gamma_G \gamma_L (M-N) EKL\sigma_G^2}{(1+\lambda) N(M-1)} \right)
$$

$$
+ \frac{\gamma_G \gamma_L E}{1+\lambda} \cdot \left\| \nabla f_{1:K-1} \left( \theta_K^0 \right) \right\|^2
$$

$$
\overset{(c_{12})}{\leq} f_{1:K} \left( \theta_K^t \right) - \frac{1}{2} \left( \frac{\gamma_G \gamma_L E}{1+\lambda} \right) \left( 1 - \frac{1}{K} \right) \left\| \nabla f_{1:K} \left( \theta_K^t \right) \right\|^2
$$

$$
+ \frac{\gamma_G \gamma_L E}{1+\lambda} \left( \frac{\gamma_L^2 E^2 L^2}{\lambda^2} + K + \frac{3\gamma_G \gamma_L (M-N) EKL}{(1+\lambda) N(M-1)} \right) B^2
$$

$$
+ \frac{\gamma_G \gamma_L E}{1+\lambda} \left( \left( 5\gamma_L^2 KEL^2 + \frac{60\gamma_G \gamma_L^3 (M-N) E^2 K L^3}{(1+\lambda) N(M-1)} \right) \left( \sigma_L^2 + 6E\sigma_G^2 \right) + \frac{3\gamma_G \gamma_L \sigma_L^2 L}{2N(1+\lambda)} \right.
$$

$$
\left. + \frac{(K-1)^2 E}{K} \cdot \frac{3\gamma_G \gamma_L L}{1+\lambda} \left( \frac{1}{2} + \frac{4(M-N)}{N(M-1)} \right) \sigma_T^2 + \frac{12\gamma_G \gamma_L (M-N) EKL\sigma_G^2}{(1+\lambda) N(M-1)} \right)
$$

$$
+ \frac{\gamma_G \gamma_L E}{1+\lambda} \cdot \left\| \nabla f_{1:K-1} \left( \theta_K^0 \right) \right\|^2, \tag{60}
$$

where $(c_{12})$ holds if $\frac{1}{2} - 30K\gamma_L^2 L^2 E^2 - \frac{3\gamma_G \gamma_L (M-N) L}{(1+\lambda) N(M-1)} \cdot \left( 120ME^3 L^2 K \gamma_L^2 + 4MEK \right) \geq 0$ and $\gamma_G \leq \frac{1}{K-1}$.

Rearranging (60) and summing it from $t = 0$ to $T - 1$, we have:

$$\sum_{t=0}^{T-1} \left(1 - \frac{1}{K}\right) \frac{\gamma_G \gamma_L E}{2(1+\lambda)} \mathbb{E} \left\|\nabla f_{1:K}\left(\theta_K^t\right)\right\|^2 \leq f_{1:K}\left(\theta_K^0\right) - f_{1:K}\left(\theta_K^T\right)$$

$$+ \frac{\gamma_G \gamma_L E}{1+\lambda} \left(\frac{\gamma_L^2 E^2 L^2}{\lambda^2} + K + \frac{3\gamma_G \gamma_L (M-N) EKL}{(1+\lambda) N (M-1)}\right) B^2$$

$$+ \frac{\gamma_G \gamma_L E}{1+\lambda} \left(\left(5\gamma_L^2 KEL^2 + \frac{60\gamma_G \gamma_L^3 (M-N) E^2 KL^3}{(1+\lambda) N (M-1)}\right)\left(\sigma_L^2 + 6E\sigma_G^2\right) + \frac{3\gamma_G \gamma_L \sigma_L^2 L}{2N(1+\lambda)}\right.$$

$$+ \frac{(K-1)^2 E}{K} \cdot \frac{3\gamma_G \gamma_L L}{1+\lambda} \left(\frac{1}{2} + \frac{4(M-N)}{N(M-1)}\right) \sigma_T^2 + \left.\frac{12\gamma_G \gamma_L (M-N) EKL\sigma_G^2}{(1+\lambda) N (M-1)}\right)$$

$$+ \frac{\gamma_G \gamma_L E}{1+\lambda} \cdot \left\|\nabla f_{1:K-1}\left(\theta_K^0\right)\right\|^2, \tag{61}$$

which implies,

$$\min_{t \in [T]} \mathbb{E} \left\|\nabla f_{1:K}\left(\theta_K^t\right)\right\|^2 \leq \frac{f_{1:K}^0 - f_{1:K}^*}{\frac{1 - \frac{1}{K}}{2(1+\lambda)} E\gamma_G \gamma_L T} + \Psi,$$

where

$$\Psi = \frac{2}{1 - \frac{1}{K}} \left[\left(\frac{\gamma_L^2 E^2 L^2}{\lambda^2} + K + \frac{3\gamma_G \gamma_L (M-N) EKL}{(1+\lambda) N (M-1)}\right) B^2\right.$$

$$+ \left(5\gamma_L^2 KEL^2 + \frac{60\gamma_G \gamma_L^3 (M-N) E^2 KL^3}{(1+\lambda) N (M-1)}\right)\left(\sigma_L^2 + 6E\sigma_G^2\right) + \frac{3\gamma_G \gamma_L \sigma_L^2 L}{2N(1+\lambda)}$$

$$+ \frac{(K-1)^2 E}{K} \cdot \frac{3\gamma_G \gamma_L L}{1+\lambda} \left(\frac{1}{2} + \frac{4(M-N)}{N(M-1)}\right) \sigma_T^2 + \frac{12\gamma_G \gamma_L (M-N) EKL\sigma_G^2}{(1+\lambda) N (M-1)}$$

$$+ \left.\left\|\nabla f_{1:K-1}\left(\theta_K^0\right)\right\|^2\right].$$

$\square$

# F ADDITIONAL EXPERIMENTAL DETAILS

## F.1 DATASET.

We conduct our experiments on the popular domain shift datasets:

- **Digit-10**: It contains 10 digit categories and consists of 4 datasets: **MNIST** LeCun et al. (2010), **USPS** Hull (2002), **SVHN** Netzer et al. (2011), and **EMINIST** Cohen et al. (2017). MNIST and EMNIST are handwritten style digits, but they are from different sources. SVHN is a real-world digit dataset from street view hour numbers, and the images in USPS are collected by the U.S. Postal Service. It contains 380,548 images in the training set and 78,039 images in the testing set.

- **VLCS** Torralba & Efros (2011): It contains 5 categories and consists of 4 datasets: VOC2007, LabelMe, Caltech-101, and Sun09. The domains differ in picture style, background and are taken with different shooting parameters. There are 7,486 images in all this dataset.

- **PACS** Li et al. (2017): It contains 7 categories and the four domains are photo, art painting, cartoon, and sketch. The variation across domains lies in the image style. PACS contains a total of 9,991 images.

- **DN4IL** Gowda et al. (2023): It is a is a subset of the DomainNet dataset, and it contains 100 categories and consists of 6 domains: clipart, infograph, painting, quickdraw, real, sketch. Since the original DomainNet dataset contains 345 redundancy classes, the DN4IL subtract the most representative and significant classes with size of 100 to be more effective for continual learning algorithms to train and test the capability.

## F.2 Hyperparameters.

Our experiments are consist of two parts: the main experiment and the ablation study experiment.

- The main experiment focuses on implementing SPECIAL in the three datasets and comparing it with baselines in two metrics, i.e., ACC and BWT. In the main experiment, the local learning rate $\gamma_L$ is initialized as $0.001$ and decayed with $0.96$ after each $5$ epochs, the global learning rate $\gamma_G$ is initialized as $1$ at the first task and decayed as $\frac{1}{i}$ at task $i$. In addition, we consider the total number of workers to be $8$ and the number of participated clients to be $4$, and we set communication rounds $T = \{20/20/30/30\}$ for $\{\text{Digit-10/VLCS/PACS/DN4IL}\}$ and local epochs $E = 5$ for every dataset to guarantee the model converges in each task.

- In the ablation study experiment, the research focus has shifted to compare the performance of SPECIAL in different hyperparameter setting, we keep all settings consistent with the main experiment. We conduct ablation experiments with varying parameters in Digit-10 to observe the effect: we set communication round $T = \{1, 5, 10, 20, 30\}$, local epoch $E = \{1, 3, 5, 10\}$, and Dirichlet level $\alpha = \{0.05, 0.1, 0.3, 0.5, 0.1, 100\}$. We follow manual grid search to estimate the effects of different regularization coefficients $\lambda$ in all three datasets, using a step size of $0.2$ in the range $[0, 1]$, and further narrow the step size between the two best-performing parameters to search the best coefficient. The result shows that the best coefficient for $\{\text{Digit-10/VLCS/PACS}\}$ is between $\{[0.2, 0.4]/[0.4, 0.6]/[0, 0.2]\}$, and we further estimate the performance on $\{0.25, 0.30, 0.35\}/\{0.45, 0.50, 0.55\}/\{0.05, 0.10, 0.15\}/\{0.05, 0.10, 0.15\}$ to get the best coefficient as $\{0.25/0.40/0.05/0.15\}$.

## F.3 Experiment Details of "Client- vs. Server-side Proximal Terms"

In Section 2.2, we compare the performance of models with client-side and server-side proximal term to explore the effect of different locations of the proximal term, and the model with client-side proximal term is denoted as SPECIAL-C in the following sections. The observation is obtained from the model performance in the training process on the second task of Digit-10.

After obtaining the parameter result of each communication round, we estimate the result on three types of test datasets while training on task 2:

- the first task is recognized as the previous task, so the test dataset of task 1 is used to estimate the memorization ability of two models;

- the second task is recognized as the current task, so the test dataset of task 2 is used to measure the ability of learning new information with different settings;

- the combination of the two test datasets is used to estimate the overall performance.

The difference in the term location leads to the distinct update rules between SPECIAL and SPECIAL-C. In the task $i, i \geq 2$, the local update rule for client $m$ in each epoch is

$$\theta_{i,m}^{t,m} = \text{prox}_\lambda \left( \theta_{i,m}^{t,e} - \gamma_G g_{i,m}^{t,e} \right),$$

where $\text{prox}_\lambda (x) := \arg\min_{\theta \in \mathbb{R}^d} \left\{ \frac{1}{2} \|\theta - x\|^2 + \lambda \|\theta - \theta_{i-1}\|^2 \right\}$ is the proximal mapping Xiao & Zhang (2014). After collecting the difference vectors, the server of SPECIAL-C averages them without incorporating the previous global parameters. The complete algorithm of SPECIAL-C is summarized in Algorithm 2.

# G  The Use of Large Language Models (LLMs)

During the preparation of this work, we used large language models solely to assist with polishing the writing.

**Algorithm 2** The SPECIAL-C Algorithm

---

1: Initialize $\theta_0$
2: **for** task $i = 1$ to $K$ **do**
3:   Set task initial point: $\theta_i^0 = \theta_{i-1}$
4:   **for** round $t = 0$ to $T - 1$ **do**
5:     The server randomly selects a set $\mathcal{S}_i^t$ of $N$ clients
6:     **for** client $m \in \mathcal{S}_i^t$ **in parallel do**
7:       Download from server: $\theta_{i,m}^{t,0} = \theta_i^t$
8:       **for** epoch $e = 0$ to $E - 1$ **do**
9:         Calculate: $g_{i,m}^{t,e} = \nabla f_{i,m}\left(\theta_{i,m}^{t,e}, \xi_{i,m}^{t,e}\right)$
10:        Local update: $\theta_{i,m}^{t,e+1} = \begin{cases} \theta_{i,m}^{t,e} - \gamma_L g_{i,m}^{t,e}, & i = 1 \\ \theta_{i,m}^{t,m} = \text{prox}_\lambda\left(\theta_{i,m}^{t,e} - \gamma_L g_{i,e}^{t,e}\right), & i \geq 2 \end{cases}$
11:      **end for**
12:      $\Delta_{i,m}^t = \theta_{i,m}^{t,E} - \theta_{i,m}^{t,0} = -\gamma_L \sum_{e=0}^{E-1} g_{i,m}^{t,e}$
13:      Send $\Delta_{i,m}^t$ to server
14:     **end for**
15:     Server receives all $\Delta_{i,m}^t$
16:     Compute: $\Delta_i^t = \frac{1}{M} \sum_{m=1}^{M} \Delta_{i,m}^t$
17:     Server update: $\theta_i^{t+1} = \theta_i^t + \gamma_G \Delta_i^t$
18:   **end for**
19:   Set final model of task $i$: $\theta_i = \theta_i^T$
20: **end for**

---

