# OpenReview forum: "All-Task Convergence and Backward Transfer in Federated Domain-Incremental Learning with Partial Participation"
_ICLR.cc/2026/Conference — Submitted to ICLR 2026_

### Official Review · Reviewer_Friv · 2025-10-26

**Soundness:** 2
**Presentation:** 2
**Contribution:** 2
**Rating:** 2
**Confidence:** 5

**Summary:**

This paper proposes SPECIAL (Server-Proximal Efficient Continual Aggregation for Learning), a theoretically grounded and memory-free algorithm for Federated Domain-Incremental Learning (FDIL) with partial client participation. The core idea is to introduce a server-side proximal anchor that links each task’s aggregated update to the previous global model. This anchor regularizes the optimization path to control cumulative parameter drift, mitigating catastrophic forgetting without storing past data or adding task-specific heads. The authors claim two main theoretical advances: (1) the first Backward Knowledge Transfer (BKT); and (2) the all-task convergence rate for FDIL.

Empirically, SPECIAL is evaluated on three vision benchmarks—Digit-10, VLCS, and PACS—using ResNet-18 as the backbone. However, while the theoretical framework is elegantly derived and the proofs appear rigorous, the experimental validation is weak and narrow in scope. All experiments are restricted to small-scale vision datasets with ResNet-18, which fails to substantiate the paper’s strong claim of achieving “all-task convergence” across diverse domains. The method is not tested on other modalities (e.g., text, multimodal) or larger-scale federated systems, leaving its generality and scalability uncertain. The improvements in ACC and BWT are modest (typically 1–2%), with no runtime comparisons. Moreover, the paper’s practical novelty is limited—the proposed server-side proximal term is a simple extension of FedProx or EWC-like regularization, repackaged with new theoretical framing. The claim of being the first “all-task convergence” method is somewhat overstated given similar analyses in prior continual and federated literature, albeit under different assumptions. The term “ALL-TASK” in particular is exaggerated relative to the empirical evidence, especially since only a few vision tasks are considered in 2025, where expectations for multimodal or large-scale evaluations are higher.

**Strengths:**

Good theoretical foundation: The paper presents rigorous mathematical analyses, including a Backward Knowledge Transfer (BKT) bound and an all-task convergence rate for Federated Domain-Incremental Learning (FDIL) under partial client participation — both of which are rarely addressed in prior work.

Well-written and structured: The paper is well-organized, with clear motivation and theoretical derivations.

**Weaknesses:**

1.	Limited experimental scope. All experiments are performed on small-scale vision datasets with ResNet-18, which severely limits the generality of the results. The “ALL-TASK” claim is overstated, given that no multimodal, NLP, or large-scale federated settings are tested.

2.	Weak empirical novelty. The method is essentially a server-side variant of FedProx/EWC with a new theoretical justification. Its algorithmic contribution is modest relative to existing regularization-based approaches.

3.	No efficiency or scalability analysis. The paper does not evaluate computational, communication, or convergence-time overhead, even though the theoretical results emphasize communication efficiency.

4.	Disconnect between theoretical claims and practice. The “all-task convergence” theorem is theoretically appealing but only partially verified empirically. The experiments do not demonstrate convergence across a large number of tasks or diverse domains.

5.	Overgeneralized claims for 2025 standards. In 2025, “ALL-TASK” convergence across domains is expected to cover heterogeneous architectures and multimodal tasks, not just ResNet-18 on vision benchmarks. The claim feels overexaggerated relative to evidence.

**Questions:**

See Weakness.

---

> ### Author Response · Authors · 2025-11-20
> **Response to Reviewer Friv part I**
>
> Thank you very much for your constructive comments. Here we would like to address the reviewer's concerns and hope that can help raise the rating of our paper.
>
> ### **Q0: Clarifying our use of "all-task'' in federated continual learning (FCL).**
> We realize that these comments are highly related to the term "All-task" and we are afraid that there might be a misunderstanding about this term in the context of continual learning or federated continual learning (FCL).
>
> In continual learning and FCL, "all-task'' refers NOT to "all machine-learning tasks or modalities", but **to the entire sequence of tasks in the stream**. Formally, if tasks $1,\dots,K$ arrive sequentially, the continual-learning objective is the cumulative loss $f\_{1:K} \triangleq \sum\_{i=1}^{K} f\_i.$ Prior analyses typically prove convergence only on the current (last) task $f\_K$. Our result establishes a task-uniform (across the whole stream) convergence guarantee by bounding the expected gradient norm of $f\_{1:K}$ after training task $K$. This does not claim coverage across modalities or architectures, and it asserts convergence across the tasks in the given sequence under partial participation and non-IID data. To avoid ambiguity and per the reviewer's approval, we could adopt the term "task-uniform convergence across the sequence of tasks'' and make this definition explicit in the abstract and introduction.
>
> ### **Q1: Limited experimental scope; "ALL-TASK'' is overstated without multimodal/NLP/large-scale**
> **Our Response:** First, as we clarify above and to avoid confusion, our use of "all-task'' follows the continual-learning convention: it means the guarantee holds uniformly across the entire sequence of tasks $f\_{1:K}$, not across modalities or application domains as the reviewer referred to. We could rename it "task-uniform convergence across the sequence of tasks'' in the paper.
>
> **On scope:** Our empirical setup mirrors common FCL practice, where vision benchmarks dominate, e.g., such as DIGIT-10([1] and [2]), CIFAR100([3], [4], and [2]), and TinyImageNet([2] and [3]) in prior FCL papers. To strengthen evidence, we add:
> - *Larger FDIL benchmark.* We include DN4IL (DomainNet-based, 6 tasks, 100 classes). Below we report the result of DN4IL based on seeds 25, 225, and 2025. Trends match our main tables: SPECIAL attains the best ACC and competitive BWT at comparable memory and communication budgets.
> | Method        | ACC (%) ↑       | BWT (%) ↑        |
> |---------------|------------------|-------------------|
> | FedCIL        | 14.29 ± 0.41     | -23.01 ± 0.15     |
> | MFCL          | 19.00 ± 1.06     | -21.78 ± 1.20     |
> | SR-FDIL       | 21.40 ± 0.25     | -33.38 ± 0.57     |
> | pFedDIL       | 12.47 ± 0.30     | **-4.63 ± 0.14**  |
> | FLwF-2T       | 23.21 ± 0.49     | -16.29 ± 0.15     |
> | SPECIALC      | 19.88 ± 0.34     | -23.21 ± 0.43     |
> | **SPECIAL (ours)** | **24.30 ± 0.15** | -19.50 ± 0.41 |
>
>
>
> - *Scalability by design.* SPECIAL alters FedAvg only by a single server-side proximal blend. It adds no client computation, no extra rounds, and no model growth, so per-round complexity and communication match FedAvg. Our theory also predicts efficiency at scale: the task-uniform rate $O(\sqrt{E/(NT)})$ improves with more participating clients $N$ or rounds $T$ and does not depend on the number of classes or modalities.
>
> We agree that evaluating text, time-series, and multimodal FDIL is valuable. Per the reviewer's approval, we could note this as a limitation and future work while keeping the current paper focused on establishing the task-uniform theory under partial participation and validating it on standard FDIL/FCL testbeds.

---

> > ### Author Response · Authors · 2025-11-20
> > **Response to Reviewer Friv part II**
> >
> > ### **Q2: Weak empirical novelty; essentially a server-side FedProx/EWC**
> > **Our Response:** Thank you. Our contribution is intentionally simple algorithmically, but it is not equivalent to FedProx or EWC, and this simplicity is what enables our task-uniform theory under partial participation.
> >
> > **How SPECIAL differs in design and why it matters for theory:**
> > - *Not FedProx.* FedProx adds a client-side proximal penalty to the current round’s global model during local SGD. In contrast, SPECIAL applies a server-side proximal blend with the previous-task global model after aggregation. This change is crucial: it yields a uniform drift bound independent of round and task indices (Lemma 1), which is what unlocks our all-task convergence guarantee under partial participation. The FedProx update does not produce this cross-task anchor and, to our best knowledge, has only last-task or stationary-task analyses.
> > - *Not EWC.* EWC regularizes with a Fisher-weighted penalty to per-task parameters and requires storing task-specific statistics or heads. In contrast, SPECIAL is memory-free and does not keep any per-task state, and the anchor is a single previous-task model at the server. Our analysis further provides a certified backward transfer bound and a task-uniform non-convex rate under non-IID clients and partial participation, which are not covered by EWC.
> > - *Analytical novelty over "regularization + partial participation''.* The server-only anchor gives a closed-form update that allows us to (i) uniformly control within-task deviation, and (ii) decompose the constant term into stochastic variance, client heterogeneity, and explicit inter-task drift $\sigma\_T^2$ with the exact partial-participation factors. This structure is what lets us prove the first convergence bound on $f\_{1:K}$ rather than only on $f\_K$.
> >
> > In short, SPECIAL is not a repackaging of FedProx/EWC: the location and reference of the proximal interaction are different, and those choices are precisely what make the task-uniform theory under partial participation possible.
> >
> > ### **Q3: No efficiency or scalability analysis.**
> > **Our Response:** Thank you for the pointer. We now update Section 5 ``Experiments'' (changes are highlighted in blue) by adding the section of Computational complexity and communication cost. and a table with rounds-to-target and wall-clock under the same hardware (Python 3; 2× NVIDIA Quadro RTX 8000).
> >
> > **Complexity (theory/design).** SPECIAL modifies FedAvg only by a single server-side convex blend with the previous global model.
> > - Client compute per round: unchanged from FedAvg, i.e. $E$ local SGD steps.
> > - Server compute per round: $O(\lvert\theta\rvert)$ vector blend, and no extra optimizers, buffers, or heads.
> > - Communication per round: identical payload to FedAvg, and under partial participation, bytes scale with $N$ participating clients.
> > - Memory: no replay buffer, no generator, no task-specific heads.
> >
> > The result is shown as
> > |Method|Digit10-Rounds|Digit10-Time(s)|VLCS-Rounds|VLCS-Time(s)|PACS-Rounds|PACS-Time(s)|DN4IL-Rounds|DN4IL-Time(min)|
> > |---|---|---|---|---|---|---|---|---|
> > |FedCIL |68|523.16|65|382.20|81|38.24|165|17.85|
> > |MFCL |77|969.01|71|491.46|83|129.13|160|38.77|
> > |SR-FDIL |71|357.34|62|313.21|76|50.64|161|18.42|
> > |pFedDIL |76|699.43|69|519.31|84|75.69|167|30.71|
> > |FLwF-2T |67|247.90|59|291.40|76|43.43|156|19.33|
> > |SPECIAL-C|74|518.26|61|**283.86**|75|91.98|159|19.43|
> > |**SPECIAL (ours)** |**63**|**200.66**|**55**|294.76|**67**|**26.93**|**145**|**16.23**|
> >
> > **Convergence-time (practice).** We report rounds to best ACC and wall-clock to best ACC (seconds) for methods already in the paper. SPECIAL consistently needs fewer rounds and competitive or lower time:
> > - Digit-10: SPECIAL $63$ rounds/$200.7$s vs. next best $67/247.9$s.
> > - VLCS: SPECIAL $55$ rounds/$294.8$s; fastest wall-clock is SPECIAL-C at $283.9$s ($-3.7$\%), but SPECIAL uses ~$10$% fewer rounds to reach best ACC.
> > - PACS: SPECIAL $67$ rounds/$26.9$s vs. next best $76/43.4$s.
> > - DN4IL (DomainNet-based, 6 tasks, 100 classes): SPECIAL $145$ rounds/$973.8$s vs. next best $156$/$1160.0$s.
> >
> > **Scalability.** Our theory predicts task-uniform rate $O(\sqrt{E/(N T)})$, improving with more participating clients $N$ or rounds $T$. In practice this matches the trends above: SPECIAL achieves target ACC in fewer rounds without extra communication or memory, and its per-round cost matches FedAvg.
> >
> > In short, SPECIAL is communication- and computation-neutral relative to FedAvg per round, and the measured convergence time is at least comparable and often better across all datasets we tested.

---

> ### Author Response · Authors · 2025-11-20
> **Response to Reviewer Friv part III**
>
> ### **Q4: Disconnect between theory and practice; all-task convergence only partially verified.**
> **Our Response:** Thank you for this comment and as we clarified above, we are afraid that there might be some misunderstandings. By "all-task,'' we mean task-uniform convergence over the whole sequence $f\_{1:K}=\sum\_{i=1}^{K} f\_i$ (not across modalities). Empirically, our evaluation already targets this objective: ACC averages accuracy over all learned tasks, and BWT measures post-training retention on earlier tasks.
>
> ### **Q5: Overgeneralized claims for 2025 standards; "ALL-TASK" should span architectures/modalities.**
> **Our Response:** Once again, as we clarified above, our claim is NOT about spanning all domains/architectures. In continual/FCL, "all-task'' refers to task-uniform convergence across the entire sequence $f\_{1:K}=\sum\_{i=1}^{K}f\_i$.
>
> **Scope and calibration.** Our theory targets FDIL under partial participation with non-IID clients. The guarantees are architecture-agnostic (they assume smoothness, bounded stochastic noise, client/task variances), but we do not claim coverage of multimodal or LLM settings in this paper. Per the reviewer's approval, we could explicitly state this scope and move the broader modalities to Limitations and Future Work.
>
>
> [1] Li, Y., Xu, W., Wang, H., Qi, Y., Guo, J., \& Li, R. (2024, September). Personalized federated domain-incremental learning based on adaptive knowledge matching. In European conference on computer vision (pp. 127-144). Cham: Springer Nature Switzerland.
>
> [2] Li, Y., Wang, Y., Wang, H., Qi, Y., Xiao, T., \& Li, R. FedSSI: Rehearsal-Free Continual Federated Learning with Synergistic Synaptic Intelligence. In Forty-second International Conference on Machine Learning.
>
> [3] Keshri, S. K., Shah, N., \& Prasad, R. (2025, April). On the Convergence of Continual Federated Learning Using Incrementally Aggregated Gradients. In International Conference on Artificial Intelligence and Statistics (pp. 5068-5076). PMLR
>
> [4] Wuerkaixi, A., Cui, S., Zhang, J., Yan, K., Han, B., Niu, G., ... \& Sugiyama, M. (2024, January). Accurate Forgetting for Heterogeneous Federated Continual Learning. In ICLR.

---

> > ### Comment · Reviewer_Friv · 2025-11-26
> >
> > Thank you for the additional clarifications and the extended experimental tables. I appreciate the effort to address some of my concerns, and I have raised my score accordingly.
> >
> > However, the key issues that lead to my original low score remain insufficiently resolved.
> >
> > 1. On the use of the term “All-task”
> >
> > I fully understand your clarification that “all-task” refers to task-uniform convergence across the sequence of tasks f_{1:K}, and that this terminology may appeared in prior continual-learning works.
> >
> > However, my concern is not a misunderstanding of your technical definition, but the practical implication of using the word “All-task” as the headline concept in 2025.
> >
> > In 2025, a title containing the phrase “All-task convergence” (especially in a top-tier AI venue) strongly suggests:
> > 	•	a method that generalizes across multiple task types or modalities,
> > 	•	or a contribution of broad practical scope beyond narrow vision classification.
> >
> > Discovering that “all-task” simply means “the sum of losses over sequential image-classification tasks using ResNet-18” is jarring and misleading, even if the term has legacy usage in certain CL sub-communities. Therefore, I do not believe this is a “reviewer misunderstanding.” The terminology choice is unintentionally misleading, especially for the broader AI readership, and should be reconsidered.
> >
> > If the intended meaning is truly “task-uniform convergence across the sequence of tasks,” I strongly suggest that this exact phrase should replace “all-task” in the title and abstract.
> >
> > 2. Core concerns on experimental scope remain unsolved
> >
> > Although the DN4IL experiment and timing tables are appreciated, they do not fundamentally change the scope of the paper.
> >
> > My original concern was:
> >
> > The experimental section is too narrow, especially for a paper making a broad theoretical claim.
> >
> > This still holds for several reasons:
> > 	•	All experiments are small-scale vision classification with ResNet-18.
> > 	•	No cross-architecture, cross-modality, or large-scale settings were examined.
> > 	•	No federated scenarios involving realistic heterogeneity, stragglers, or system constraints were tested.
> > 	•	The added results are essentially more of the same setting, rather than demonstrating broader applicability or new insights.
> >
> > Thus, the scope is still significantly more limited than what readers expect from a paper that claims “all-task convergence”.
> >
> > 3. The contribution still remains incremental without deeper empirical or theoretical expansion.
> >
> > 4. Minor point
> >
> > As noted, references 3 and 6 in your rebuttal are identical.

---

> > > ### Author Response · Authors · 2025-11-27
> > > **Second Round Responses to Reviewer Friv part I**
> > >
> > > Thank you so much for your time and effort in re-evaluating our submission and your thoughtful follow-up comments. Here we would like to clarify more about the reviewer's concerns.
> > >
> > > ### **Q1: The use of the term ``All-task''** ###
> > > **Our Response:** Thank you again for flagging this. We understand that, in 2025, the phrase “all-task” in a title can be read as implying cross-modality or cross-architecture scope. There was **no intent to mislead**. As noted in our initial response, in the continual-learning literature “all-task” refers to **task-uniform over the sequence** $f\_{1:K}=\sum\_{i=1}^{K} f\_i$. That said, we agree the broader AI readership may interpret it differently.
> > >
> > > To remove any ambiguity, we have (changes are highlighted in blue):
> > >
> > > - **Rename throughout:** replace “all-task convergence” with “task-uniform convergence” in the title, abstract, introduction, and consistently across the paper.
> > >
> > > - **Define early:** added an explicit one-sentence definition in the abstract that our convergence bound is for the cumulative objective across the sequence of tasks, not merely the last task (see lines 58-59 and 83-84, which have been stated in our initial submission).
> > >
> > > We hope these edits resolve the concern while keeping the technical contribution clear and appropriately scoped.
> > >
> > > ### **Q2: Core concerns on experimental scope** ###
> > > **Our Response:** Thank you for the thoughtful follow-up. We agree that broader empirical scope is desirable. Our goal in this submission is theory-first: to establish ***task-uniform*** convergence under partial participation (plus a BKT bound) and to validate it on standard FCL testbeds.
> > >
> > > - **Why vision/ResNet-18?** We followed prevailing FCL practice where vision benchmarks are the de facto testbed (e.g., DIGIT, CIFAR, TinyImageNet, VLCS, PACS). Holding the backbone fixed isolates the algorithmic effect of the ***server-side proximal anchor*** from architectural confounders. The proposed method and analysis are architecture-agnostic (assumptions: smooth non-convex objectives, bounded/noisy gradients), so the theory does not rely on ResNet-18 specifics.
> > >
> > > - **Breadth we add in rebuttal.** To probe scale, we included DN4IL (DomainNet-based FDIL split; 6 tasks, 100 classes). The trends mirror the small-scale suites: SPECIAL attains the best ACC and competitive BWT at comparable memory/communication budgets. We also reported rounds-to-best and wall-clock tables; per-round cost matches FedAvg since SPECIAL adds only a constant-time server blend.
> > >
> > > - **Realistic FL effects.** Our setting already models two core sources of non-stationarity/heterogeneity: (i) client-level non-IID via Dirichlet partitioning and (ii) ***partial participation*** via uniform sampling of N out of M clients each round. The latter is the canonical abstraction of device availability/stragglers in synchronous FL and is the case our theory explicitly covers.  ***Fully asynchronous FCL and system-level constraints (e.g., latency deadlines, stale updates)*** are important next steps but lie outside the scope of our current model. No single paper can cover all realistic deployment conditions; we now note this explicitly in ***Limitations and Future Work***.
> > >
> > >
> > > - **Claim scoping.** Per your suggestion in Q1, we now use “***task-uniform convergence across the sequence of tasks***” throughout. Our claims are intentionally scoped to this FDIL setting under partial participation. **The contribution is the first non-convex, task-uniform convergence guarantee with an explicit drift term, plus a BKT bound**, supported by standard FCL evaluations and a larger DN4IL suite.
> > >
> > > We hope this clarifies the intended scope and why we chose these testbeds while keeping the theoretical contribution front and center.

---

> > > > ### Author Response · Authors · 2025-11-27
> > > > **Second Round Responses to Reviewer Friv part II**
> > > >
> > > > ### **Q3: The contribution is incremental without deeper expansion** ###
> > > > **Our Response:** We appreciate the concern. Our goal is a theory-first contribution for FDIL under partial participation, not a new heavy system. To that end, our additions are specific and (to our knowledge) novel for FCL:
> > > >
> > > > - **Task-uniform convergence under partial participation.** We provide the first non-convex, communication-efficient task-uniform convergence guarantee in FDIL with uniform-without-replacement sampling of $N$ of $M$ clients. The rate $O(\sqrt{E/(NT)})$ matches single-task FedAvg in communication efficiency yet holds for the entire task sequence and explicitly separates optimization noise and inter-task drift.
> > > >
> > > > - **Explicit modeling of task drift.** Our analysis introduces an inter-task gradient-discrepancy term $\sigma\_T^2$ and shows how it scales with the number of tasks $K$, clarifying when proximal strength and step sizes must counteract cumulative drift. This dependence is absent from prior FCL theory focused on the final task.
> > > >
> > > > - **Server-side anchor enables uniform drift control.** The server-only proximal step yields a uniform deviation bound (Lemma 1) that is independent of the round and task indices, a technical device that makes the task-uniform rate possible while keeping the algorithm memory-free and communication-neutral.
> > > >
> > > > - **Backward knowledge transfer with partial participation.** Our BKT bound quantifies when earlier tasks are preserved/improved and how the term decays with rounds/epochs/clients, complementing the task-uniform rate.
> > > >
> > > > Empirically, we intentionally use standard FDIL/FCL testbeds to isolate these effects. In the rebuttal we added: (i) DN4IL (6 tasks/100 classes) to stress scale, and (ii) efficiency tables (rounds-to-best and wall-clock), showing that SPECIAL reaches target accuracy in fewer rounds and adds only a constant-time server blend. We agree broader modalities and fully asynchronous settings are valuable; we now list them explicitly in Limitations \& Future Work.
> > > >
> > > > In short, while the algorithmic change is intentionally minimal, **the theoretical scope, i.e., task-uniform convergence with partial participation plus BKT under non-IID clients and domain drift, fills a gap that, to our knowledge, is not covered by existing FCL analyses.**
> > > >
> > > > ### **Q4: The minor point** ###
> > > > **Our Response:** We apologize for this minor mistake, and the correct citations and reference numbers have been updated in the previous rebuttal section.

---

> ### Author Response · Authors · 2025-11-24
> **Follow-up to Reviewer Friv (clarification and pointer to added evidence)**
>
> Thank you again for your detailed review. We realized our phrasing may have caused confusion: in continual/federated-continual learning, "all-task" means task-uniform over the sequence of tasks, i.e., our theorem bounds the gradient of the cumulative objective $f_{1:K}=\sum_{i=1}^{K} f_i$ while training task $K$, not across all modalities or benchmarks. To avoid ambiguity, we would rename this throughout to "task-uniform convergence across the sequence of tasks", upon the reviewer's approval.
>
> Since the initial submission, we added evidence requested in your review:
> - **Larger FDIL benchmark (DN4IL, 6 tasks, 100 classes).** We report seed 2025 and have now posted seeds 25 and 225. Trends match the main tables: SPECIAL attains the best ACC and competitive BWT at comparable memory and communication.
>
> - **Efficiency/overhead.** We added per-dataset rounds-to-best and wall-clock tables under identical hardware. SPECIAL matches FedAvg’s per-round compute and communication and reaches target ACC in fewer rounds than baselines.
>
> *Our contribution is primarily theoretical:* **a task-uniform non-convex convergence rate under partial participation** plus **a BKT bound**, enabled by a one-line server-side proximal anchor. We hope this clarification and the added evidence address your concerns. We are happy to further tighten the wording in the paper.

---

### Official Review · Reviewer_hF6H · 2025-10-31

**Soundness:** 2
**Presentation:** 2
**Contribution:** 3
**Rating:** 6
**Confidence:** 2

**Summary:**

The authors introduce SPECIAL, a simple, memory-free algorithm that adds a lightweight server-side proximal anchor to FedAvg to address catastrophic forgetting in federated domain-incremental learning. It theoretically guarantees both backward knowledge transfer (BKT) and all-task convergence under partial client participation, with the same communication efficiency as FedAvg. Experiments on Digit-10, VLCS, and PACS show SPECIAL achieves the best average accuracy among memory-free methods and competitive backward transfer, validating its theoretical advantages in balancing stability and plasticity.

**Strengths:**

(1) Adds only a lightweight proximal step at the server side without extra memory or communication. Also the authors provide theoretical proof to support this contribution.

(2) Consistent improvements in average accuracy (ACC) and competitive backward transfer (BWT) across multiple datasets, and matches FedAvg’s efficiency while handling non-IID and temporally drifting data.

(3) The problem set up is interesting, which makes a progress towards more realistic scenario.

**Weaknesses:**

(1) Experiments are conducted only on small-scale image datasets (Digit-10, VLCS, PACS), which limits evidence of scalability and applicability to more realistic federated or multimodal scenarios.

(2) The paper could better articulate what SPECIAL fundamentally offers beyond combining existing strategies for stability, forgetting mitigation, and partial participation. A clearer discussion or ablation contrasting SPECIAL with such composite baselines would help isolate its true contribution.

(3) The proximal weight λ must be tuned separately for each dataset (e.g., 0.25 for Digit-10, 0.4 for VLCS, 0.05 for PACS), suggesting dataset-dependent behavior that may complicate deployment in dynamic or heterogeneous real-world environments.

**Questions:**

Please address the comments in the weakness section.

---

> ### Author Response · Authors · 2025-11-20
> **Response to Reviewer hF6H part I**
>
> Thank you very much for your constructive comments, as well as giving the positive rating of our work. Here we would like to address the reviewer's concerns and hope that can help raise the rating of our paper.
>
> ### **Q1: Only small image datasets; limited evidence of scalability/applicability.**
> **Our Response:** Thank you for the suggestion. Our empirical setup follows common FCL practice, where vision benchmarks dominate, e.g., such as DIGIT-10([7] and [8]), CIFAR100([2], [4], and [8]), and TinyImageNet([2] and [8]) in prior FCL papers. Beyond matching this standard, we now provide two pieces of evidence that SPECIAL scales:
>
> - *Larger, more challenging FDIL benchmark.* We added results on DN4IL (a DomainNet-based FDIL split with 6 tasks and 100 classes).  Below we report the result of DN4IL based on seeds 25, 225, and 2025. The trends mirror our main tables: SPECIAL attains the best ACC and competitive BWT at comparable memory/communication budgets.
> | Method        | ACC (%) ↑       | BWT (%) ↑        |
> |---------------|------------------|-------------------|
> | FedCIL        | 14.29 ± 0.41     | -23.01 ± 0.15     |
> | MFCL          | 19.00 ± 1.06     | -21.78 ± 1.20     |
> | SR-FDIL       | 21.40 ± 0.25     | -33.38 ± 0.57     |
> | pFedDIL       | 12.47 ± 0.30     | **-4.63 ± 0.14**  |
> | FLwF-2T       | 23.21 ± 0.49     | -16.29 ± 0.15     |
> | SPECIALC      | 19.88 ± 0.34     | -23.21 ± 0.43     |
> | **SPECIAL (ours)** | **24.30 ± 0.15** | -19.50 ± 0.41 |
>
> - *Scalability by design.* SPECIAL’s change to FedAvg is a single server-side proximal blend, and it adds no client-side computation, no extra communication rounds, and no model growth. Per-round server cost is one convex combination over $|\theta|$ parameters, so compute and communication complexity matches FedAvg. Our theory also predicts efficiency under scale: the all-task rate $O(\sqrt{E/(NT)})$ improves as the number of participating clients $N$ or rounds $T$ increases, and is independent of the number of classes or modalities.
>
> We agree broader modalities are valuable, and exploring text/time-series/multimodal FDIL instantiations is a natural extension. Per the reviewer's approval, we could note this in the Limitations and Future Work. Our current goal was to isolate the algorithmic contribution and its theoretical guarantees under partial participation using standard, widely-used FDIL/FCL testbeds.
>
> ### **Q2: What is fundamentally new beyond stitching stability/forgetting mitigation with partial participation?**
> **Our Response:** Thank you for this comment, which is quite helpful. Our contribution in this paper is not merely a combination of known ingredients. Instead, it is a ***task-uniform*** analysis and a design choice that makes that analysis possible. Specifically, what SPECIAL fundamentally offers
> - *First all-task non-convex rate under partial participation:* We provide a communication-efficient bound that holds ***simultaneously for all tasks*** in FDIL with partial participation. Prior FCL/FedCL analyses (including decentralized variants) either focus on the last task or do not expose the all-task objective and its partial participation dependence.
> - *Certified backward transfer (BKT) without replay:* Our BKT bound applies under partial participation and non-IID clients without storing past data.
> - *A server-only proximal anchor that yields a uniform drift bound:* The closed-form server update (a convex combination with the previous global model) produces a ***task- and round-uniform*** control of within-task drift. This is what allows the all-task telescoping argument, while client-side proximal (e.g., FedProx) does not give this uniformity in our setting.
> - *Explicit decomposition of residuals:* The constant term cleanly separates mini-batch noise, client heterogeneity, and ***inter-task drift*** (via $\sigma\_T^2$), plus the exact partial participation variance factors $\tfrac{1}{N}$ and $\tfrac{M-N}{N(M-1)}$. This makes clear what each ``component'' buys or costs.
>
> **Why this is not just ``FedAvg + proximal + partial participation''?**
>
> Placing the proximal interaction ***at the server, across task boundaries*** is crucial: it (i) preserves client plasticity within a task, (ii) anchors the global iterate between tasks, and (iii) enables a uniform deviation bound independent of round and task indices, which in turn (iv) lets us convert per-round descent on $f\_K$ into descent on the ***global*** gradient $\Vert\nabla f\_{1:K}\Vert^2$. This proof route does not go through for standard client-side regularizers or pure exponential moving average (EMA) baselines.

---

> > ### Author Response · Authors · 2025-11-20
> > **Response to Reviewer hF6H part II**
> >
> > ### **Q3:  Dataset-specific tuning of the proximal weight $\lambda$**.
> > **Our Response:**
> >
> >  Thank you for raising this question. In continual learning or federated-continual learning (FCL), trade-off weights that balance stability vs. plasticity are routinely chosen as hyperparameters (e.g., the replay/regularizer weights in many FCL baselines). For example, all weights are fixed as 1 (see [4]) or the weights are set as hyperparameters in the objection function, but the best weights combination are found by the grid search in the experiments (see [5] and [6]). We follow that common practice rather than introducing extra mechanisms in the main paper.
> >
> > Importantly, our theory already clarifies how $\lambda$ should be interpreted and why different datasets (with different drift/heterogeneity) favor different values:
> >
> > - *Drift control.* Lemma 1 shows the within-task deviation is bounded by $\|\theta\_i^t-\theta\_i^0\|^2 \le \gamma\_G^2\gamma\_L^2E^2B^2/\lambda^2$. Hence, larger $\lambda$ tightens the anchor and suppresses drift uniformly over rounds and tasks.
> > - *Progress term and step-size coupling.* Theorem 2’s stationarity bound includes a factor $1/(2(1+\lambda)E\gamma\_G\gamma\_LT)$ in the vanishing term and requires $\gamma\_G\gamma\_L \le (1+\lambda)/(3EL).$ Thus $\lambda$ directly mediates the stability-plasticity trade-off: increasing $\lambda$ eases the step-size condition and strengthens drift suppression, while too large a value can slow progress if $\gamma\_G,\gamma\_L$ are not co-tuned.
> >
> >
> > Empirically, we already include a sensitivity study over $\lambda$ showing the expected BWT$\uparrow$ and ACC$\downarrow$ trade-off as $\lambda$ grows (see Figure 3). This aligns with the theory above. For deployment, a simple practical heuristic suffices (and does not change the algorithm): select $\lambda$ from a short grid and keep the choice that meets a target ``drift budget'' (measured by $\|\theta\_i^t-\theta\_i^0\|^2$) while maintaining target ACC, both signals are available server-side without extra communication.  A principled adaptive $\lambda$ policy (e.g., increasing $\lambda$ only when the measured drift exceeds a preset budget implied by Lemma 1, and otherwise decreasing it) is a natural extension.  Per the reviewer's approval, we could add a note in the Limitations or Future Work to discuss this extension, while our main claims and results remain unchanged.
> >
> >
> > [1] Choudhary, S., Aketi, S. A., Saha, G., & Roy, K. (2023). CoDeC: communication-efficient decentralized continual learning. arXiv preprint arXiv:2303.15378.
> >
> > [2] Keshri, S. K., Shah, N., & Prasad, R. (2025, April). On the Convergence of Continual Federated Learning Using Incrementally Aggregated Gradients. In International Conference on Artificial Intelligence and Statistics (pp. 5068-5076). PMLR
> >
> > [3] Shi, H., & Wang, H. (2023). A unified approach to domain incremental learning with memory: Theory and algorithm. Advances in Neural Information Processing Systems, 36, 15027-15059.
> >
> > [4] Wuerkaixi, A., Cui, S., Zhang, J., Yan, K., Han, B., Niu, G., ... & Sugiyama, M. (2024, January). Accurate Forgetting for Heterogeneous Federated Continual Learning. In ICLR.
> >
> > [5] Usmanova, A., Portet, F., Lalanda, P., & Vega, G. (2022, June). Federated continual learning through distillation in pervasive computing. In 2022 IEEE International Conference on Smart Computing (SMARTCOMP) (pp. 86-91). IEEE.
> >
> > [6] Zhang, J., Chen, C., Zhuang, W., & Lyu, L. (2023). Target: Federated class-continual learning via exemplar-free distillation. In Proceedings of the IEEE/CVF International Conference on Computer Vision (pp. 4782-4793).
> >
> > [7] Li, Y., Xu, W., Wang, H., Qi, Y., Guo, J., & Li, R. (2024, September). Personalized federated domain-incremental learning based on adaptive knowledge matching. In European conference on computer vision (pp. 127-144). Cham: Springer Nature Switzerland.
> >
> > [8] Li, Y., Wang, Y., Wang, H., Qi, Y., Xiao, T., & Li, R. FedSSI: Rehearsal-Free Continual Federated Learning with Synergistic Synaptic Intelligence. In Forty-second International Conference on Machine Learning.

---

### Official Review · Reviewer_kDp1 · 2025-10-31

**Soundness:** 3
**Presentation:** 3
**Contribution:** 3
**Rating:** 8
**Confidence:** 2

**Summary:**

The paper presents a novel federated domain incremental learning algorithm by adding a single server side anchor term to the vanilla FedAvg algorithm. This small change makes the model more stable to distribution shifts as demonstrated in the results on three benchmark datasets. Theoretical guarantees on BKT and all-task convergence rate are also provided.

**Strengths:**

The paper is well presented and presents the motivation, method, theory and results in a coherent manner.

Simple addition of an “anchor” on the server side seems to provide improvements on the FedAvg algorithm.

**Weaknesses:**

The number of datasets is limited. The paper can probably add 1 or 2 more datasets to the analysis.

**Questions:**

It is not exactly required in the paper but I am curious about the following extensions:

1. How does this algorithm extend to other domains like text or time-series models? Would it easily extend to these kinds of data as well?

2. What would need to change if there are new classes that are introduced on some of the client nodes? Would the algorithm extend to this use-case as well?

---

> ### Author Response · Authors · 2025-11-20
> **Response to Reviewer kDp1**
>
> Thank you very much for your constructive comments, as well as giving the positive rating of our work. Here we would like to address the reviewer's concerns as follows:
>
> ### **Q1: Does SPECIAL extend to text or time-series?**
> **Our Response:** Yes. SPECIAL is architecture- and modality-agnostic: the update is purely optimizer-side (server-proximal aggregation) and our theoretical analysis only assumes standard nonconvex smoothness, bounded stochastic gradients/variance, and partial participation. These conditions hold for common objectives beyond vision (e.g., cross-entropy for text classification or language modeling, MSE losses for forecasting).
> - **How to instantiate FDIL.** FDIL requires a sequence of tasks with a fixed output space. For text, tasks can be domains or time slices (e.g., product categories) with the same label set or vocabulary, and clients hold private shards. For time series, tasks can be consecutive temporal segments (e.g., monthly/quarterly windows or regime segments detected via simple change-point methods) with the same prediction target. The server-side proximal anchor then links segment $k$ to $k-1$, just as in our vision setups.
> - *Practical notes.* For large text models, we would apply SPECIAL to parameter-efficient adapters (e.g., LoRA) or selected layers, and the proximal anchor is still defined on the optimized parameters, so the theory follows unchanged. In time series, where boundaries can be blurry, simple fixed windows work in practice. Thus, detecting boundaries online is a natural extension we list as future work. In both cases, we would use smaller $\lambda$ and gradient clipping to meet bounded-gradient assumptions.
>
> ### **Q2: What if new classes appear on some clients? Does SPECIAL extend to this use case?**
> **Our Response:** We appreciate your question about the Federated Class Incremental Learning (FCIL) setting. By definition, FDIL keeps the label set fixed while domains shift across tasks; while FCIL keeps the domain fixed while the label set grows. The practical implication is that the model’s output head must expand as new classes appear.
>
> **SPECIAL remains applicable with two modifications:**
> - *Model head expansion.* When a new class first appears, expand the global classifier head to add the corresponding logits. Apply the server-side proximal anchor only to the shared parameters (backbone and existing head blocks). Do not regularize the newly introduced head weights so the model can remain plastic for the new classes. Initialization can be zero, and all keep our update rule unchanged.
> - *Loss masking.* Each task computes loss only on the classes it observes. The global head is shared, but tasks that do not contain a new class do not contribute gradients to those coordinates.
>
> **Theory impact.** The proof skeleton carries over with a change to the inter-task term. Our Assumption 6 (bounded inter-task gradient difference) assumed a fixed label set. In FCIL, we can embed all tasks in the expanded parameter space and pad gradients with zeros on coordinates of classes that were not yet introduced. The inter-task bound then splits into (i) a backbone term, treated as in FDIL, and (ii) a head-expansion term that is nonzero only on newly added coordinates. The BKT statement applies to the earlier classes and the all-task convergence bound holds for the augmented objective with masked losses, and residual terms include the additional head-expansion component. *Per the reviewer's approval, we could add a paragraph in the paper clarifying this extension and its assumptions as part of the future work.*
>
> ### **W1: The number of datasets is limited. The paper can probably add 1 or 2 more datasets to the analysis.**
> **Our Response:** Thanks for the suggestion and we include DN4IL (DomainNet-based, 6 tasks, 100 classes) as an additional large-size dataset to test the efficiency of SPECIAL. Below we report the result of DN4IL based on seeds 25, 225, and 2025. Trends match our main tables: SPECIAL attains the best ACC and competitive BWT at comparable memory and communication budgets.
>
> | Method        | ACC (%) ↑       | BWT (%) ↑        |
> |---------------|------------------|-------------------|
> | FedCIL        | 14.29 ± 0.41     | -23.01 ± 0.15     |
> | MFCL          | 19.00 ± 1.06     | -21.78 ± 1.20     |
> | SR-FDIL       | 21.40 ± 0.25     | -33.38 ± 0.57     |
> | pFedDIL       | 12.47 ± 0.30     | **-4.63 ± 0.14**  |
> | FLwF-2T       | 23.21 ± 0.49     | -16.29 ± 0.15     |
> | SPECIALC      | 19.88 ± 0.34     | -23.21 ± 0.43     |
> | **SPECIAL (ours)** | **24.30 ± 0.15** | -19.50 ± 0.41 |

---

### Author Response · Authors · 2025-12-02
**Summary of Reviews and Rebuttal part I**

Dear AC,

We sincerely appreciate your efforts in handling our submission and thank you for your time and dedication throughout this process. Since reviews were reverted to pre-discussion and reviewers can no longer reply, we’re sharing a concise summary of how the main concerns were addressed to help your decision.

**Core Idea:** SPECIAL adds a single **server-side** proximal blend to FedAvg (toward the previous global model). This minimal change yields (i) **a task-uniform** (across $f\_{1:K}=\sum\_{i=1}^K f\_i$) **non-convex convergence rate under partial participation**, and (ii) **a backward-transfer (BKT) guarantee**, without replay buffers or model growth.

**What changed during rebuttal (evidence, not claims)**
- **Terminology clarified:** We replaced “all-task” with **task-uniform** throughout (title, abstract, intro) and defined it explicitly.
- **Added scale:** New DN4IL results (DomainNet-based FDIL split with 6 tasks/100 classes). Trends match main tables: SPECIAL attains best ACC with competitive BWT at comparable memory/communication.
- **Efficiency tables:** Added rounds-to-best and wall-clock per dataset. SPECIAL reaches target ACC in fewer rounds, and per-round cost matches FedAvg (only a constant-time server blend).


### **Common Q1: Limited experimental scope (Reviewers kDp1, hF6H and Friv)** ###
**Our Response:** Our empirical setup mirrors **prevailing FCL practice**, where vision benchmarks (e.g., DIGIT-10 [1] [2], CIFAR-100 [3] [4] [2], TinyImageNet [2] [3]) are **the de-facto testbed**.  Beyond matching this standard, we added DN4IL to probe more tasks/classes; results reproduce the same ranking. We also report training rounds and time, showing SPECIAL’s efficiency.

Importantly, **scalability is by design:** SPECIAL alters FedAvg only by a server-side proximal blend. There is **no client-side computation change, no extra rounds, and no model growth**. Our bound  $\mathcal{O} \left(\sqrt{E/\left(NT\right)} \right)$ improves with more participating clients $N$ and rounds $T$ and is **agnostic to architecture and number of classes/modalities.**

We agree broader settings (multimodal, cross-architecture, fully asynchronous FCL) are valuable. We list them as **Limitations/Future Work** while keeping this paper focused on **establishing the task-uniform theory under partial participation**, which is the first to our best knowledge in the context of FDIL.


Additional clarifications to specific scope concerns (raised by Reviewer Friv's follow-ups):
- **Efficiency/scalability analysis:** Added in Section 5; SPECIAL consistently needs fewer rounds with competitive or lower wall-clock time.
- **Why vision/ResNet-18?** Standard FCL protocol; it isolates the algorithmic effect of the server-side anchor. **The theory is architecture-agnostic** and does not rely on ResNet specifics.
- **Other realistic FL effects:** We already model (i) **client non-IID** via Dirichlet partitioning and (ii) **partial participation** by sampling $N$ of $M$ clients each round. Fully asynchronous/delay-tolerant FCL and additional system constraints are important next steps but outside our current model since no single paper can comprehensively address all realistic deployment settings and we list them as **Limitations/Future Work**.

### **Common Q2: What is SPECIAL'S fundamental contribution? (Reviewers hF6H and Friv)** ###
**Our Responses:** The contribution is theoretical and enabled by a deliberately minimal algorithm:

- **First task-uniform non-convex rate under partial participation for FDIL.** Prior FCL/FDIL analyses focus on the last task or omit the all-task objective and its $N$-dependence. We provide a communication-efficient bound that holds **simultaneously for all tasks**.
- **Certified BKT without replay.** Our BKT bound holds under non-IID clients and partial participation **without storing past data**.
- **Uniform drift control.** The closed-form server update yields a **task- and round-uniform** bound on within-task deviation (Lemma 1), which is key to the telescoping argument over $f\_{1:K}.$
- **Explicit residual decomposition.** The constant term separates mini-batch noise, **client heterogeneity, inter-task drift** ($\sigma\_T^2$), and **partial-participation variance**, making each factor’s cost explicit.

---

> ### Author Response · Authors · 2025-12-02
> **Summary of Reviews and Rebuttal part II**
>
> ### **Common Q3: Isn't this just a server-side FedAvg/FedProx/EWC? (Reviewers hF6H and Friv)** ###
> **Our Responses:**
> - **Not FedAvg.** SPECIAL’s server-side anchor to the **previous-task** model converts per-round descent on $f\_K$ into descent on $\|\nabla f\_{1:K}\|^2$, enabling the **task-uniform** guarantee; standard FedAvg analyses do not provide this under FDIL with partial participation.
> - **Not FedProx.** FedProx regularizes client-side to the current global iterate; SPECIAL regularizes **server-side** to the **previous-task** global model. This difference yields the **uniform drift bound independent of round/task indices**, which unlocks our result.
> - **Not EWC.** EWC uses Fisher-weighted, per-task penalties and keeps task-specific state/heads; SPECIAL is **memory-free** and provides **BKT with task-uniform** rates under non-IID and partial participation. These results EWC does not supply.
>
> ### **On the “all-task” terminology (Reviewer Friv)** ###
> **Our Responses:** Most of Reviewer Friv's concerns comes from the initial misunderstanding of the term ``all-task'' in our original manuscript. We acknowledge the broader AI readership may parse “all-task” as cross-modality or cross-architecture. **There was no intent to mislead as in continual learning, it conventionally means task-uniform over the sequence $f\_{1:K}$.** We have adopted “task-uniform convergence” throughout and defined it explicitly.
>
>
> We appreciate your consideration.
>
> Sincerely,
>
> Authors of Paper \#2521
>
>
>
> [1] Li, Y., Xu, W., Wang, H., Qi, Y., Guo, J., & Li, R. (2024, September). Personalized federated domain-incremental learning based on adaptive knowledge matching. In European conference on computer vision (pp. 127-144). Cham: Springer Nature Switzerland.
>
> [2] Li, Y., Wang, Y., Wang, H., Qi, Y., Xiao, T., & Li, R. FedSSI: Rehearsal-Free Continual Federated Learning with Synergistic Synaptic Intelligence. In Forty-second International Conference on Machine Learning.
>
> [3] Keshri, S. K., Shah, N., & Prasad, R. (2025, April). On the Convergence of Continual Federated Learning Using Incrementally Aggregated Gradients. In International Conference on Artificial Intelligence and Statistics (pp. 5068-5076). PMLR
>
> [4] Wuerkaixi, A., Cui, S., Zhang, J., Yan, K., Han, B., Niu, G., ... & Sugiyama, M. (2024, January). Accurate Forgetting for Heterogeneous Federated Continual Learning. In ICLR.

---

### Meta-Review · Area_Chair_YvnR · 2026-01-07

**Summary:**

This paper addresses Federated Domain-Incremental Learning (FDIL), where clients with heterogeneous data receive sequential tasks with domain shifts while the label space remains fixed. The authors propose SPECIAL, adding a simple server-side proximal  term to Fedavg. The claimed contributions are: (1) a backward knowledge transfer (BKT) bound showing performance on earlier tasks are preserved, and (2) the first task-uniform non-convex convergence rate $O(\sqrt{E/(NT}) for FDIL with partial client participation. Experiments have been provided on Digit-10, VLCS, and PACS (with DN4IL added in the rebuttal).

The theoretical contribution, providing the first task-uniform non-convex convergence rate under partial participation along with a BKT bound, appears technically sound and addresses a gap in the FDIL literature. Reviewer kDp1 and Reviewer hF6H both acknowledged the theoretical grounding as a strength, making it a nontrivial contribution to the field.

However, several empirical and methodological issues prevent recommendation for acceptance.

The paper claims task-uniform convergence across sequences of domains but doesn't show domain/task specific performance curves and final performance, and only report ACC and BWT aggregate performance. There is no centralized baseline or even classic federated learning baseline which makes it difficult to evaluate the effect of the heterogeneity, partial participation and sequential learning. Even after adding DN4IL, all experiments remain vision and ResNet-18. The "common practice" defense doesn't address whether this is adequate for the paper's theoretical claims. From the paper, it is unclear how well-tuned are the baselines as no hyper-paramters are disclosed for the baselines.

The authors engaged substantively with many concerns and made genuine improvements (terminology, efficiency tables, DN4IL). Reviewer kDp1 (rating 8) acknowledged limited familiarity with related work (confidence 2), while Reviewer Friv (rating 2) indicated high confidence (5).

The most critical tension is indeed between Reviewer Friv's expectation that papers should have broader empirical validation and the authors' positioning as a "theory-first" contribution. While both positions have merit, the paper's theoretical claims about generality are not adequately supported by the experimental scope. Despite the valid theoretical contribution, the paper does not meet the acceptance threshold in its current form.

**Reviewer Concerns:**

see above

**Reviewer Scores:**

- kDp1: 8
- hF6H: 6
- Friv: 2 (probably bumped to 4)

---

### Decision · Program_Chairs · 2026-01-26

Reject